# Are Vision Transformers Robust to Spurious Correlations?

## Abstract

Deep neural networks may be susceptible to learning spurious correlations that hold on average but not in atypical test samples. As with the recent emergence of vision transformer (ViT) models, it remains unexplored how spurious correlations are manifested in such architectures. In this paper, we systematically investigate the robustness of different transformer architectures to spurious correlations on three challenging benchmark datasets and compare their performance with popular CNNs. Our study reveals that for transformers, larger models and more pre-training data significantly improve robustness to spurious correlations. Key to their success is the ability to generalize better from the examples where spurious correlations do not hold. Further, we perform extensive ablations and experiments to understand the role of the self-attention mechanism in providing robustness under spuriously correlated environments. We hope that our work will inspire future research on further understanding the robustness of ViT models to spurious correlations.

## 1 Introduction

A key challenge in building robust image classification models is the existence of *spurious correlations*: misleading heuristics imbibed within the training dataset that are correlated with majority examples but do not hold in general. Prior works have shown that convolutional neural networks (CNNs) can rely on spurious features to achieve high average test accuracy. Yet, such models lead to low accuracy on rare and untypical test samples lacking those heuristics (Sagawa et al., 2020a; Geirhos et al., 2019; Goel et al., 2021; Tu et al., 2020). In Figure 1, we illustrate a model setup that exploits the spurious correlation between the `water background` and label `waterbird` for prediction. Consequently, a model that relies on spurious features can perform poorly on test samples where the correlation no longer holds, such as `waterbird` on `land background`. Recent work by Sagawa et al. (2020b) has shown that over-parameterization in CNNs exacerbates robustness to spurious correlations.

As with the paradigm shift to attention-based architectures, it becomes increasingly critical to understand their behavior under ill-conditioned data. Particularly, in this paper, we provide a first study on the following question: *Are Vision Transformers robust to spurious correlations*? Although the behavior of ViTs against occlusions, perturbations, and distributional shifts have been extensively studied in the literature (Bhojanapalli et al., 2021; Bai et al., 2021; Zhang et al., 2022; Paul & Chen, 2022; Park & Kim, 2022), it remains unexplored how spurious correlation is manifested in the recent development of vision transformers. Unlike prior works, we specifically focus on robustness performance on challenging datasets designed to expose spurious correlations learned by the model. The learning task is challenging as the model is more likely to learn those spurious associations while training to achieve high training accuracy and fail on those samples where the spurious associations do not hold at test time. In contrast, previous works on ViT robustness did not consider this problem and focused on robustness to perturbations and data shifts during inference. Thus, the nature of the "robustness" problem studied in our paper is fundamentally different from the prior art.

Motivated by the question, we systematically investigate how and when transformer-based models exhibit robustness to spurious correlations on challenging benchmarks. We base our findings after studying a variety of transformer architectures including ViT (Dosovitskiy et al., 2021), Swin transformer (Liu et al., 2021b), Pyramid Vision Transformer (Wang et al., 2021) and recently proposed Robust Vision Transformer (RVT) (Shi et al., 2022). Our findings reveal that: (1) For transformers, larger models and more pre-training data

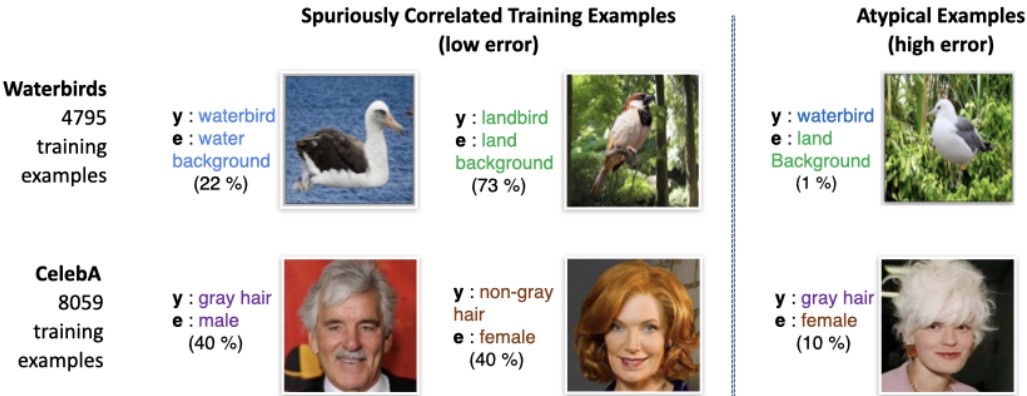

Figure 1: **Representative Examples.** We visualize samples from Waterbirds (Sagawa et al., 2020a) and CelebA (Liu et al., 2015) dataset. The label $y$ is spuriously correlated with environment $e$ in the majority of training samples. The frequency of each group in training data is denoted by (%). Figure is adapted from (Sagawa et al., 2020a).

yield a significant improvement in robustness to spurious correlations. On the other hand, when pre-trained on a relatively smaller dataset such as ImageNet-1k, transformer-based models have a higher propensity of memorizing training samples and are less robust to spurious correlations compared to the ImageNet-21k counterpart. (2) Interestingly, our linear probing experiment shows that the improvement in robustness does not simply stem from strong pre-trained features. (3) The key reason for success can be attributed to the ability to generalize better from those examples where spurious correlations do not hold while fine-tuning. However, despite better generalization capability, ViT models suffer high errors on challenging benchmarks when these counterexamples are scarce in the training set. (4) Finally, our study also reveals that, under the small pre-training regime, modern transformer architectures (Mao et al., 2022) despite being designed to be robust against common corruptions, out-of-distribution and adversarial attacks perform poorly when fine-tuned on datasets containing spurious associations further highlighting the importance of this study.

Going beyond, we perform extensive ablations to understand the role of the self-attention mechanism in providing robustness to ViT models. Our findings reveal that the self-attention mechanism in ViTs plays a crucial role in guiding the model to focus on spatial locations in an image which are essential for accurately predicting the target label. To the best of our knowledge, we provide a first systematic study on the robustness of Vision Transformers when learned on datasets containing spurious correlations. Our key contributions are summarized below:

1. Our work sheds light on the impact of the pre-training dataset and model capacity on ViT's robustness to spurious correlations. We observe that larger models and more pre-training data improve model robustness to spurious correlations.

2. We perform extensive experiments and ablations to understand the impact of linear probing, fine-tuning, data imbalance, etc., on model robustness. These ablations lead to previously unknown findings on ViT's robustness to spurious correlations.

3. We provide insights on ViT's robustness by analyzing the attention matrix, which encapsulates essential information about the interaction among image patches.

4. We design systematic experiments to disentangle the effect of different data augmentation and regularizations on robustness to spurious correlations for both transformer and CNN models.

We hope our work will inspire future research on further understanding the robustness of ViT models when fine-tuned on datasets containing spurious correlations.

## 2    Related Works

**Pre-training and robustness.**    Recently, there has been an increasing amount of interest in studying the effect of pre-training (Kolesnikov et al., 2020; Devlin et al., 2019; Radford et al., 2021; Liu et al., 2019; Shi et al., 2022). Specifically, when the target dataset is small, generalization can be significantly improved through pre-training and then finetuning (Zeiler & Fergus, 2014). Findings of Hendrycks et al. (2019) reveal that pre-training provides significant improvement to model robustness against label corruption, class imbalance, adversarial examples, out-of-distribution detection, and confidence calibration. Recent work (Liu et al., 2021a) has also shown that self-supervised learning makes CNNs more immune to dataset imbalance in pre-training datasets. The study by (Taori et al., 2019) showed that in pre-trained models there exists a linear relationship between the in-distribution and out-of-distribution performance. Further, models pre-trained on larger and more diverse data are generally found to be more effectively robust (Taori et al., 2019; Andreassen et al., 2021). Recently Andreassen et al. (2021) provided a detailed investigation on the impact of model and dataset size on the effective robustness of pre-trained models. In this work, we focus distinctly on robustness to *spurious correlation*, and how it can be improved through large-scale pre-training. To the best of our knowledge, this problem has not been investigated in prior literature on pre-training.

**Vision transformer.**    Since the introduction of transformers by Vaswani et al. (2017) in 2017, there has been a deluge of studies adopting the attention-based transformer architecture for solving various problems in natural language processing (Radford et al., 2018; 2019; Yang et al., 2019; Dai et al., 2019). In the domain of computer vision, Dosovitskiy et al. (2021) first introduced the concept of Vision Transformers (ViT) by adapting the transformer architecture in Vaswani et al. (2017) for image classification tasks. Subsequent studies (Dosovitskiy et al., 2021; Steiner et al., 2021) have shown that when pre-trained on sufficiently large datasets, ViT achieves superior performance on downstream tasks, and outperforms state-of-art CNNs such as residual networks (ResNets) (He et al., 2016) of comparable sizes. Since coming to the limelight, multiple variants of ViT models have been proposed. Touvron et al. showed that it is possible to achieve comparable performance in small pre-training data regimes using extensive data augmentation and a novel distillation strategy. Further improvements on ViT include enhancement in tokenization module (Yuan et al., 2021), efficient parameterization for scalability (Touvron et al., 2021b; Xue et al., 2021; Zhai et al., 2021) and building multi-resolution feature maps on transformers (Liu et al., 2021b; Wang et al., 2021). Recent studies (Zhou et al., 2022; Mao et al., 2022) propose enhanced ViT architectures to provide additional robustness against adversarial attacks (Mao et al., 2022) and image occlusion/corruptions (Mao et al., 2022; Zhou et al., 2022) during inference. While we also consider these architectures, our focus significantly differs from prior works — *we provide a first systematic study on the robustness of vision transformers when the training data itself contains inherent biases.*

**Robustness of transformers.**    Naseer et al.  provides a comprehensive understanding of the working principle of ViT architecture through extensive experimentation. Some notable findings in Naseer et al. (2021) reveal that transformers are highly robust to severe occlusions, perturbations, and distributional shifts. Recently, the performance of ViT models in the wild has been extensively studied (Bhojanapalli et al., 2021; Zhang et al., 2022; Paul & Chen, 2022; Bai et al., 2021; Tian et al., 2022; Park & Kim, 2022), using a set of robustness generalization benchmarks, e.g., ImageNet-C (Hendrycks & Dietterich, 2019), Stylized-ImageNet (Geirhos et al., 2019), ImageNet-A (Hendrycks et al., 2021), etc. Different from prior works, we focus on robustness performance on challenging datasets, which are designed to expose spurious correlations learned by the model. Our analysis reveals that: (1) Robustness against occlusions, perturbations, and distribution shifts does not necessarily guarantee robustness against spurious correlations. (2) Large-scale pre-training helps to mitigate this problem by better generalizing on examples from under-represented groups. Our findings are also complementary to robustness studies (Tu et al., 2020; He et al.; McCoy et al., 2019) in the domain of natural language processing, which reported that transformer-based BERT (Devlin et al., 2019) models improve robustness to spurious correlations.

## 3 Preliminaries

### 3.1 Spurious Correlations

Spurious features refer to statistically informative features that work for the majority of training examples but do not capture essential cues related to the labels (Sagawa et al., 2020a; Geirhos et al., 2019; Goel et al., 2021; Tu et al., 2020). We illustrate a few examples in Figure 1. In `waterbird` vs `landbird` classification problem, majority of the training images has the target label (`waterbird` or `landbird`) spuriously correlated with the background features (`water` or `land` background). Sagawa et al. showed that deep neural networks can rely on these statistically informative yet spurious features to achieve high test accuracy on average, but fail significantly on groups where such correlations do not hold such as `waterbird` on `land background`.

Formally, we consider a training set, $\mathcal{D}^{\text{train}}$, consisting of $N$ training samples: $\{\mathbf{x}_i, y_i\}_{i=1}^N$, where samples are drawn independently from a probability distribution: $\mathcal{P}_{X,Y}$. Here, $X \in \mathcal{X}$ is a random variable defined in the pixel space, and $Y \in \mathcal{Y} = \{1, \dots, K\}$ represents its label. We further assume that the data is sampled from a set of $E$ environments $\mathcal{E} = \{e_1, e_2, \cdots, e_E\}$. The training data has spurious correlations, if the input $\mathbf{x}_i$ is generated by a combination of invariant features $\mathbf{z}_i^{inv} \in \mathbb{R}^{d_{inv}}$, which provides essential cues for accurate classification, and environmental features $\mathbf{z}_i^e \in \mathbb{R}^{d_e}$ dependent on environment $e$:

$$\mathbf{x}_i = \rho(\mathbf{z}_i^{inv}, \mathbf{z}_i^e).$$

Here $\rho$ represents a function transformation from the feature space $[\mathbf{z}_i^{inv}, \mathbf{z}_i^e]^T$ to the pixel space $\mathcal{X}$. Considering the example of `waterbird` vs `landbird` classification, invariant features $\mathbf{z}_i^{inv}$ would refer to signals which are essential for classifying $\mathbf{x}_i$ as $y_i$, such as the feather color, presence of webbed feet, and fur texture of birds, to mention a few. Environmental features $\mathbf{z}_i^e$, on the other hand, are cues not essential but correlated with target label $y_i$. For example, many waterbird images are taken in water habitats, so water scenes can be considered as $\mathbf{z}_i^e$. Under the data model, we form groups $g = (y, e) \in \mathcal{Y} \times \mathcal{E}$ that are jointly determined by the label $y$ and environment $e$. For this study, we consider the binary setting where $\mathcal{E} = \{1, -1\}$ and $\mathcal{Y} = \{1, -1\}$, resulting in four groups. The concrete meaning for each environment and label will be instantiated in corresponding tasks, which we describe in Section 4.

### 3.2 Transformers

Similar to the Transformer architecture in Vaswani et al. (2017), ViT model expects the input as a 1D sequence of token embeddings. An input image is first partitioned into non-overlapping fixed-size square patches of resolution $P \times P$, resulting in a sequence of flattened 2D patches. Following Devlin et al. (2019), ViT prepends a learnable embedding (`class` token) to the sequence of embedded patches, and this `class` token is used as image representation at the output of the transformer. To imbibe relative positional information of patches, position embeddings are further added to the patch embeddings.

The core architecture of ViT mainly consists of multiple stacked encoder blocks, where each block primarily consists of: (1) multi-headed self-attention layers, which learn and aggregate information across various spatial locations of an image by processing interactions between different patch embeddings in a sequence; and (2) a feed-forward layer. See an expansive discussion in related work (Section 2).

### 3.3 Model Zoo

In this study, we aim to understand the robustness of transformer-based models when trained on a dataset containing spurious correlations and how they fare against popular CNNs. In particular, we contrast ViT with Big Transfer (BiT) models (Kolesnikov et al., 2020) that are primarily based on the ResNet-v2 architecture. For both ViT and BiT models, we consider different variants that differ in model capacity and pre-training dataset. Specifically, we use model variants pre-trained on both ImageNet-1k (IN-1k) (Russakovsky et al., 2015) and on ImageNet-21k (IN-21k) (Deng et al., 2009) datasets. For completeness, we also provide detailed ablations on other state-of-art architectures including DeiT-III (Touvron et al., 2022), Swin transformer (Liu et al., 2021b), Robust Vision Transformer (RVT) (Mao et al., 2022), Pyramid Vision Transformer (PVT) (Wang et al., 2021), and ConvNeXt (Liu et al., 2022) models.

Table 1: Average and worst-group accuracies over train and test set for different models when finetuned on Waterbirds. Both ViT-B/16 and ViT-S/16 attain better test worst-group accuracy as compared to BiT models. All models are pre-trained on ImageNet-21k. Results (mean and std) are estimated over 3 runs for each setting.

| Model | Params | Train | | Test | |
|---|---|---|---|---|---|
| | | Average Acc. | Worst-Group Acc. | Average Acc. | Worst-Group Acc. |
| ViT-B/16 | 86.1M | 100 | 100 | $\mathbf{96.75}_{\pm 0.05}$ | $\mathbf{89.30}_{\pm 1.95}$ |
| ViT-S/16 | 21.8M | 100 | 100 | $96.30_{\pm 0.51}$ | $85.45_{\pm 1.16}$ |
| ViT-Ti/16 | 5.6M | 95.7 | 81.6 | $89.50_{\pm 0.05}$ | $71.65_{\pm 0.16}$ |
| BiT-M-R50x3 | 211M | 100 | 100 | $94.90_{\pm 0.05}$ | $80.51_{\pm 1.02}$ |
| BiT-M-R101x1 | 42.5M | 100 | 100 | $94.05_{\pm 0.07}$ | $77.50_{\pm 0.50}$ |
| BiT-M-R50x1 | 23.5M | 100 | 100 | $92.05_{\pm 0.05}$ | $75.10_{\pm 0.62}$ |

**Notation:** To indicate input patch size in ViT models, we append "/x" to model names. We prepend -B, -S, -Ti to indicate `Base`, `Small` and `Tiny` version of the corresponding architecture. For instance: ViT-B/16 implies the `Base` variant with an input patch resolution of $16 \times 16$. In this paper, we use a $16 \times 16$ input patch size for computational simplicity.

## 4 Robustness to Spurious Correlation

In this section, we systematically measure the robustness performance of ViT models when fine-tuned on datasets containing spurious correlations, and compare how their robustness fares against popular CNNs. For both ViTs and CNNs, we fine-tune on datasets starting from ImageNet-21k pre-trained models. For evaluation benchmarks, we adopt the same setting as in Sagawa et al. (2020a). Specifically, we consider the following three classification datasets to study the robustness of ViT models in a spurious correlated environment: Waterbirds (Section 4.1), CelebA (Section 4.2), and ColorMNIST. Refer to Appendix, for results on ColorMNIST.

### 4.1 Waterbirds

Introduced in Sagawa et al. (2020a), this dataset contains spurious correlation between the background features and target label $y \in \{$`waterbird`, `landbird`$\}$. The dataset is constructed by selecting bird photographs from the Caltech-UCSD Birds-200-2011 (CUB) (Wah et al., 2011) dataset and then superimposing on either of $e \in \mathcal{E} = \{$`water`, `land`$\}$ background selected from the Places dataset (Zhou et al., 2017). The spurious correlation is injected by pairing `waterbirds` on `water` background and `landbirds` on `land` background more frequently, as compared to other combinations. The dataset consists of $n = 4795$ training examples, with the smallest group size of 56 (i.e., waterbird on the land background).

**Results and insights on generalization performance.** Table 1 compares worst-group accuracies of different models when fine-tuned on Waterbirds (Sagawa et al., 2020a) using empirical risk minimization. Note that all the compared models are pre-trained on ImageNet-21k. This allows us to isolate the effect of model architectures, in particular, ViT vs. BiT models. The worst-group test accuracy reflects the model's generalization performance for groups where the correlation between the label $y$ and environment $e$ does not hold. A high worst-group accuracy is indicative of less reliance on the spurious correlation in training. Our results suggest that: (1) ViT-B/16 attains a significantly higher worst-group test accuracy (89.3%) than BiT-M-R50x3 despite having a considerably smaller capacity (86.1M vs. 211M). (2) Furthermore, these results reveal a correlation between generalization performance and model capacity. With an increase in model capacity, both ViTs and BiTs tend to generalize better, measured by both average accuracy and worst-group accuracy. The relatively poor performance of ViT-Ti/16 can be attributed to its failure to learn the intricacies within the dataset due to its compact capacity.

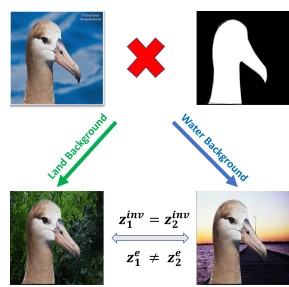 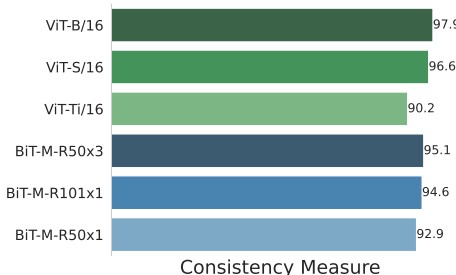

Figure 2: **Consistency Measure.** In Waterbirds dataset, $y \in \{\texttt{waterbird}, \texttt{landbird}\}$ is correlated with environment $e \in \{\texttt{water}, \texttt{land}\}$. **Left**: Visual illustration of the experimental setup for measuring model consistency. Ideally, changing the spurious features ($\mathbf{z}^e$) should have no impact on model prediction. **Right**: Evaluation results quantifying consistency for models of different architectures and varying capacity. All the models are pre-trained on ImageNet-21k.

**Results and insights on robustness performance.** We now delve deeper into the robustness of ViT models. In particular, we investigate the robustness in model prediction under varying background features. Our key idea is to compare the predictions of image pairs $(\mathbf{x}_i, \bar{\mathbf{x}}_i)$ with the same foreground object yet different background features (i.e., $\texttt{water}$ vs. $\texttt{land}$ background). We define *Consistency Measure* of a model as: $\frac{\sum_{i=1}^{N} \mathbb{I}\{\hat{f}(\mathbf{x}_i) = \hat{f}(\bar{\mathbf{x}}_i) | \hat{f}(\mathbf{x}_i) = y_i\}}{\sum_{i=1}^{N} \mathbb{I}\{\hat{f}(\mathbf{x}_i) = y_i\}}$, where $y_i$ denotes the target label. Among samples with correct predictions, this metric measures the fraction of samples with consistent predictions. To generate the image pairs $(\mathbf{x}_i, \bar{\mathbf{x}}_i)$, we first take a foreground bird photograph using the pixel-level segmentation masks from the CUB dataset (Wah et al., 2011). We then place it on the top of $\texttt{water}$ and $\texttt{land}$ background images from the Places dataset (Zhou et al., 2017). We generate multiple such pairs to form the evaluation dataset $\{(\mathbf{x}_i, \bar{\mathbf{x}}_i)\}_{i=1}^{N}$ and use this dataset to quantify the robustness performance. For this study, the evaluation dataset consists of $N = 11788$ paired samples.

Figure 2 provides a visual illustration of the experimental setup (**left**), along with the evaluation results (**right**). Our operating hypothesis is that a robust model should predict the same class label $\hat{f}(\mathbf{x}_i)$ and $\hat{f}(\bar{\mathbf{x}}_i)$ for a given pair $(\mathbf{x}_i, \bar{\mathbf{x}}_i)$, as they share exactly the same foreground object (i.e., invariant feature). Our results in Figure 2 show that ViT models achieve overall higher consistency measures than BiT counterparts. For example, the best model ViT-B/16 obtains consistent predictions for 93.9% of image pairs. Overall, using ViT pre-trained models yields strong generalization and robustness performance on Waterbirds.

### 4.2 CelebA

Beyond background spurious features, we further validate our findings on a different type of spurious feature based on gender attributes. Here, we investigate the behavior of machine learning models when learned on training samples with spurious associations between target label and demographic information such as gender. Following Ming et al. (2022), we use CelebA dataset, consisting of celebrity images with each image annotated using 40 binary attributes. We have the label space $\mathcal{Y} = \{\texttt{gray hair}, \texttt{nongray hair}\}$ and gender as the spurious feature, $\mathcal{E} = \{\texttt{male}, \texttt{female}\}$. The training data consists of 4010 images with label $\texttt{grey hair}$, out of which 3208 are $\texttt{male}$, resulting in a spurious association between gender attribute $\texttt{male}$ and label $\texttt{grey hair}$. Formally, $\mathbb{P}(e = \texttt{grey hair} | y = \texttt{male}) \approx \mathbb{P}(e = \texttt{non-grey hair} | y = \texttt{female}) \approx 0.8$.

**Results.** We see from Table 2 that ViT models achieve higher test accuracy (both average and worst-group) as opposed to BiTs. In particular, ViT-B/16 achieves $+\mathbf{4.3\%}$ higher worst-group test accuracy than BiT-M-R50x3, despite having a considerably smaller capacity (86.1M vs. 211M). These findings along with our observations in Section 4.1 demonstrate that ViTs are not only more robust when there are strong associations between the label and background features, but also avoid learning spurious correlations between demographic features and target labels.

Table 2: Average and worst-group accuracies over train and test set for different models when finetuned on CelebA. Both ViT-B/16 and ViT-S/16 attain better worst-group accuracy as compared to BiT models. All models are pre-trained on ImageNet-21k. Results (mean and std) are estimated over 3 runs for each setting.

| Model | Params | Train | | Test | |
|---|---|---|---|---|---|
| | | Average Acc. | Worst-Group Acc. | Average Acc. | Worst-Group Acc. |
| ViT-B/16 | 86.1M | 100 | 100 | $\mathbf{97.40}_{\pm 0.42}$ | $\mathbf{94.10}_{\pm 0.51}$ |
| ViT-S/16 | 21.8M | 100 | 100 | $96.26_{\pm 0.66}$ | $91.50_{\pm 1.56}$ |
| ViT-Ti/16 | 5.6M | 97.9 | 93.3 | $96.71_{\pm 0.18}$ | $88.60_{\pm 3.92}$ |
| BiT-M-R50x3 | 211M | 100 | 100 | $97.31_{\pm 0.05}$ | $89.80_{\pm 0.42}$ |
| BiT-M-R101x1 | 42.5M | 100 | 100 | $97.20_{\pm 0.08}$ | $89.33_{\pm 0.78}$ |
| BiT-M-R50x1 | 23.5M | 100 | 100 | $96.82_{\pm 1.20}$ | $87.72_{\pm 1.56}$ |

Table 3: Investigating effect of large-scale pre-training on model robustness to spurious correlations. All models are fine-tuned on Waterbirds. Pre-training on ImageNet-21k (IN-21k) provides better performance.

| | | Model | Params | FLOPs | Train Accuracy | | Test Accuracy | |
|---|---|---|---|---|---|---|---|---|
| | | | | | Average | Worst-Group | Average | Worst-Group |
| Transformer Models | IN-21k | DeiT-III-Base | 85.8M | 17.5G | 100 | 100 | 95.7 | 82.5 |
| | | DeiT-III-Medium | 38.3M | 7.5G | 100 | 100 | 94.2 | 80.8 |
| | | DeiT-III-Small | 21.8M | 4.6G | 100 | 100 | 93.6 | 76.2 |
| | IN-1k | DeiT-III-Base | 85.8M | 17.5G | 100 | 100 | 91.6 | 69.6 |
| | | DeiT-III-Medium | 38.3M | 7.5G | 100 | 100 | 93.2 | 69.8 |
| | | DeiT-III-Small | 21.8M | 4.6G | 100 | 100 | 90.6 | 65.1 |
| | IN-21k | Swin-Base | 86.7M | 15.4G | 100 | 100 | 95.7 | 87.5 |
| | | Swin-Small | 48.8M | 8.7G | 100 | 100 | 94.8 | 83.2 |
| | | Swin-Tiny | 27.5M | 4.5G | 100 | 100 | 93.9 | 78.5 |
| | IN-1k | Swin-Base | 86.7M | 15.4G | 100 | 100 | 92.3 | 61.7 |
| | | Swin-Small | 48.8M | 8.7G | 100 | 100 | 93.1 | 62.5 |
| | | Swin-Tiny | 27.5M | 4.5G | 100 | 100 | 91.3 | 50.7 |
| Convolutional Models | IN-21k | BiT-M-R50x3 | 211M | 1.4M | 100 | 100 | 94.9 | 80.5 |
| | | BiT-M-R101x1 | 42.5M | 0.5M | 100 | 100 | 94.1 | 77.5 |
| | | BiT-M-R50x1 | 23.5M | 0.4M | 100 | 100 | 92.1 | 75.1 |
| | IN-1k | BiT-S-R50x3 | 211M | 1.4M | 100 | 100 | 87.0 | 60.3 |
| | | BiT-S-R101x1 | 42.5M | 0.5M | 100 | 100 | 87.3 | 64.9 |
| | | BiT-S-R50x1 | 23.5M | 0.4M | 100 | 100 | 86.3 | 63.5 |
| | IN-21k | ConvNeXt-B | 87.5M | 15.4G | 100 | 100 | 93.1 | 76.7 |
| | | ConvNeXt-S | 49.5M | 8.7G | 100 | 100 | 92.5 | 74.2 |
| | | ConvNeXt-T | 27.9M | 4.5G | 100 | 100 | 90.3 | 69.6 |
| | IN-1k | ConvNeXt-B | 87.5M | 15.4G | 100 | 100 | 87.3 | 66.1 |
| | | ConvNeXt-S | 49.5M | 8.7G | 100 | 100 | 86.4 | 67.6 |
| | | ConvNeXt-T | 27.9M | 4.5G | 100 | 100 | 85.1 | 63.2 |

# 5 Discussion: A Closer Look at ViT Under Spurious Correlation

In this section, we perform extensive ablations to understand the role of transformer models under spurious correlations. For consistency, we present the analyses below based on the Waterbirds dataset.

### 5.1 How does the size of the pre-training dataset affect robustness to spurious correlations?

In this section, we aim to understand the role of large-scale pre-training on the model's robustness to spurious correlations. Specifically, we compare pre-trained models of different capacities, architectures, and sizes of pre-training data. To understand the importance of the pre-training dataset, we compare models pre-trained on ImageNet-1k (1.3 million images) and ImageNet-21k (12.8 million images). We report results for different transformer- and convolution-based models in Table 3. Specifically, we report for DeiT-III (Touvron et al., 2022) & Swin (Liu et al., 2021b) models from the transformer family and BiT (Kolesnikov et al., 2020) & ConvNeXt (Liu et al., 2022) architectures based on convolutions. For detailed ablation results on other transformer architectures: RVT (Mao et al., 2022) and PVT (Wang et al., 2021), readers can refer to Table 10 (Appendix). Based on the results in Table 3, we highlight the following observations:

1. First, large-scale pre-training improves the performance of the models on challenging benchmarks. For transformers, larger models (`base` and `medium`) and more pre-training data (ImageNet-21k) yield a significant improvement in all reported metrics. Hence, larger pre-training data and increasing model size play a crucial role in improving model robustness to spurious correlations. We also see a similar trend in the convolution-based model family: BiT (Kolesnikov et al., 2020) and ConvNeXt (Liu et al., 2022) models.

2. Second, when pre-trained on a relatively smaller dataset such as ImageNet-1k, both vision transformers and convolution-based models have a higher propensity of memorizing training samples and are less robust to spurious correlations compared to the ImageNet-21k counterpart. In particular, previous work (Sagawa et al., 2020b) theoretically showed that overparameterization in CNNs—increasing model size well beyond the point of zero training error—can hurt test error in minority groups. Our study further provides new insights by showing that this trend of memorizing training samples can be alleviated using large-scale pre-training data. In the extreme case without any pre-training, we show in Appendix J that both transformer and CNN models severely overfit the training dataset to attain 100% training accuracy and significantly fail on test samples.

### 5.2 Is linear probing sufficient for robustness to spurious correlation?

One may hypothesize that the robustness to spurious correlation can benefit from the strong pre-trained features. To verify this, we conduct the "linear probing" experiments by freezing all the learned parameters from pre-training, and only training a linear classifier on top. Note that this preserves the information entirely from pre-training. From Table 4, we observe that the model fails to learn the essential cues necessary for accurate classification on the Waterbirds (Sagawa et al., 2020a) dataset. This is evident from both poor training and test accuracy over the worst-group samples (where spurious correlations do not hold). Interestingly, simply preserving the pre-training distribution does not sufficiently provide ViT's robustness to spurious correlation. Further, Table 10 (Appendix) summarizes the robustness of RVT (Mao et al., 2022) models when fine-tuned on Waterbirds. Note, RVT architecture is specifically modified to provide additional robustness against adversarial attacks, common corruptions, and out-of-distribution inputs. Our results indicate that despite these additional architectural modifications, ImageNet-1k pre-trained RVT models perform poorly when fine-tuned on Waterbirds, indicating a lack of robustness to spurious correlations. We believe our work will trigger a new round of thinking in the community about the problem of spurious correlations and ways to mitigate it, from an architectural design perspective.

### 5.3 Understanding role of self-attention mechanism for improved robustness in ViT models

Given the results above, a natural question arises: what makes ViT particularly robust in the presence of spurious correlations? In this section, we aim to understand the role of ViT by looking into the self-attention mechanism. The attention matrix in ViT models encapsulates crucial information about the interaction between different image patches.

**Latent pattern in attention matrix.** To gain insights, we start by analyzing the attention matrix, where each element in the matrix $a_{i,j}$ represents attention values with which an image patch $i$ focuses on another patch $j$. For example: consider an input image of size $384 \times 384$ and patch resolution of $16 \times 16$, then we

Table 4: Investigating the effect of linear probing pre-trained models on Waterbirds (Sagawa et al., 2020a).

| | Model | Params | Train Accuracy | | Test Accuracy | |
|---|---|---|---|---|---|---|
| | | | Average | Worst-Group | Average | Worst-Group |
| IN-21k | ViT-B/16 | 86.1M | 98.6 | 80.3 | 89.5 | 68.3 |
| | ViT-S/16 | 21.8M | 95.6 | 39.3 | 86.2 | 35.9 |
| | ViT-Ti/16 | 5.6M | 88.5 | 13.2 | 82.3 | 0.1 |
| IN-21k | BiT-M-R50x3 | 211M | 100 | 100 | 90.5 | 68.3 |
| | BiT-M-R101x1 | 42.5M | 99.9 | 98.2 | 88.5 | 64.9 |
| | BiT-M-R50x1 | 23.5M | 99.8 | 98.0 | 88.6 | 62.1 |

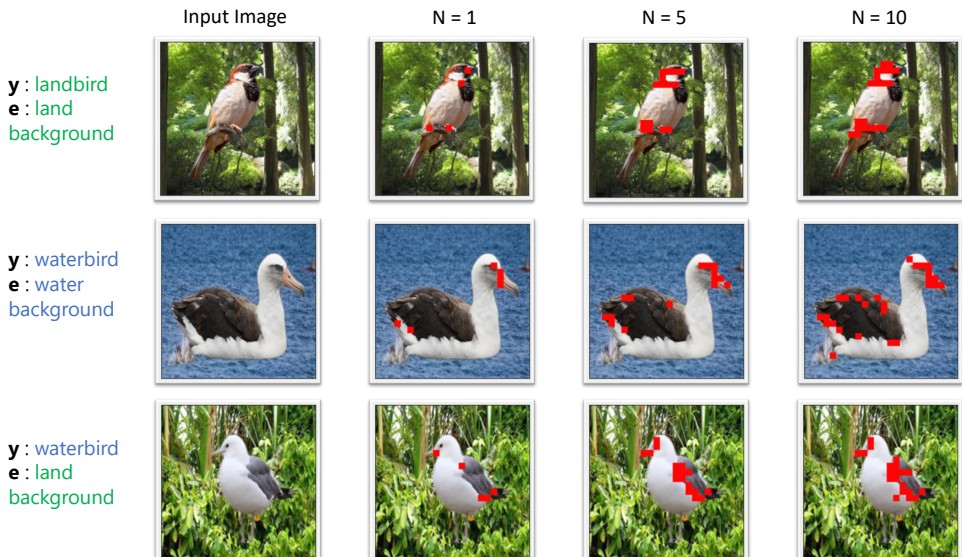

Figure 3: Visualization of the top $N$ patches receiving the highest attention (marked in red). Investigating the attention matrix, we find that all image patches—irrespective of spatial location— provide maximum attention to the patches representing essential cues for accurately identifying the foreground object such as claw, beak, and fur color. See text for details. See the Appendix for visualizations of other datasets and models.

have a $576 \times 576$ attention matrix (excluding the `class token`). To compute the final attention matrix, we use Attention Rollout (Abnar & Zuidema, 2020) which recursively multiplies attention weight matrices in all layers below. Our analysis here is based on the ViT-B/16 model fine-tuned on Waterbirds. Intriguingly, we observe that each image patch, irrespective of its spatial location, provides maximum attention to the patches representing essential cues for accurately identifying the foreground object.

Figure 3 exhibits this interesting pattern, where we mark (in red) the top $N = \{1, 5, 10\}$ patches being attended by every image patch. To do so, for every image patch $i$, where $i \in \{1, \cdots, 576\}$, we find the top $N$ patches receiving the highest attention values and mark (in red) on the original input image. This would give us $576 \times N$ patches, which we overlay on the original image. Note that different patches may share the same top patches, hence we observe a sparse pattern. In Figure 3, we can see that the patches receiving the highest attention represent important signals such as the shape of the beak, claw, and fur color—all of which are essential for the classification task `waterbird` vs `landbird`. It is particularly interesting to note the last row in Figure 3, which is an example from the minority group (`waterbird` on `land` background). This is a challenging case where the spurious correlations between $y$ and $e$ do not hold. A non-robust model would utilize the background environmental features for predictions. In contrast, we notice that each patch in the image correctly attends to the foreground patches.

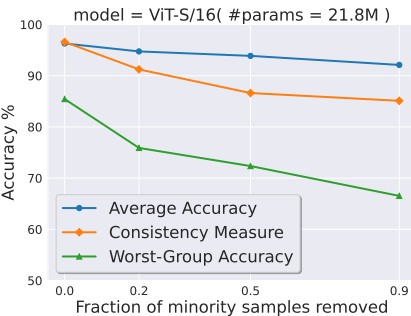 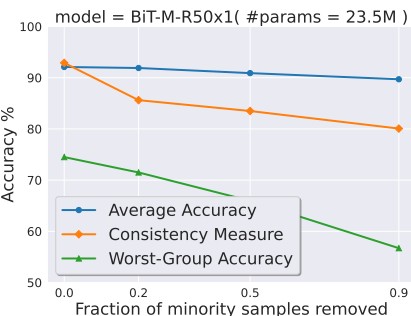

Figure 4: **Data Imbalance.** We investigate the effect of data imbalance on different model architectures. Our findings reveal that both ViT and BiT models suffer from spurious correlations when minority samples are scarce in the fine-tuning dataset.

We also provide observations for ViT-Ti/16 model in Appendix H. Specifically, for ViT-Ti/16, we notice: (1) The model correctly attends to patches responsible for accurate classification in images belonging to the majority groups, i.e., `waterbird` on `water` background and `landbird` on `land` background. (2) For images belonging to minority groups such as `waterbird` on `land` background, the model provides maximum attention to the environmental features, exhibiting a lack of robustness to spurious correlations. These experiments further help us observe a direct correlation between model robustness and patches being attended.

### 5.4  Investigating model performance under data imbalance

Recall that model robustness to spurious correlations is correlated with its ability to generalize from the training examples where spurious correlations do not hold. We hypothesize that this generalization ability varies depending on inherent data imbalance. In this section, we investigate the effect of data imbalance. In an extreme case, the model only observes 5 samples from the underrepresented group.

**Setup.**  Considering the problem of `waterbird` vs `landbird` classification, these examples correspond to those in the groups: `waterbird` on `land` background and `landbird` on `water` background. We refer to these examples that do not include spurious associations with label as minority samples. For this study, we remove varying fractions of minority samples from the smallest group (`waterbird` on `land` background), while fine-tuning. We measure the effect based on the worst-group test accuracy and model consistency defined in Section 4.1.

**Takeaways.**  In Figure 4, we report results for ViT-S/16 and BiT-M-R50x1 model when finetuned on Waterbirds dataset (Sagawa et al., 2020a). We find that as more minority samples are removed, there is a graceful degradation in the generalization capability of both ViT and BiT models. However, the decline is more prominent in BiTs with the model performance reaching near-random when we remove 90% of minority samples. From this experiment, we observe that for a fixed fraction of minority samples, ViT-S/16 attains higher worst-group accuracy than BiT-M-R50x1 model. Thus, we conclude that the additional robustness of ViT models to spurious associations stems from their better generalization capability from minority samples. However, both ViTs and BiTs still suffer from spurious correlations when minority examples are scarce.

### 5.5  Does longer fine-tuning in ViT improve robustness to spurious correlations?

Recent studies in the domain of natural language processing (Tu et al., 2020; Zhang et al., 2020) have shown that the performance of BERT (Devlin et al., 2019) models on smaller datasets can be significantly improved through longer fine-tuning. In this section, we investigate if longer fine-tuning also plays a positive role in the performance of ViT models in spuriously correlated environments.

**Takeaways** Figure 5 reports the loss (**left**) and accuracy (**right**) at each epoch for ViT-S/16 model fine-tuned on Waterbirds dataset (Sagawa et al., 2020a). To better understand the effect of longer fine-tuning on worst-group accuracy, we separately plot the model loss and accuracy on all examples and minority samples. From the loss curve, we observe that the training loss for minority examples decreases at a much slower

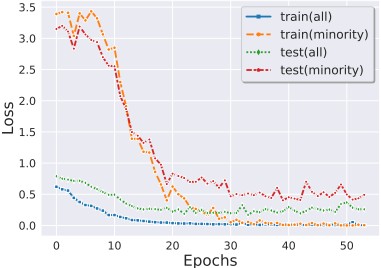 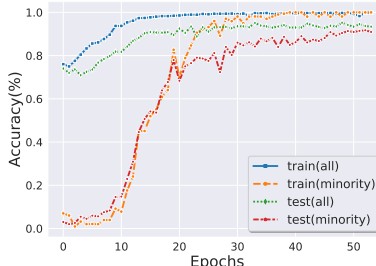

Figure 5: **Longer Fine-tuning.** We study the effect of longer fine-tuning on the performance of ViT models. We report loss and accuracy for ViT-S/16 model finetuned on Waterbirds (Sagawa et al., 2020a) at each epoch of fine-tuning. Investigating further we observe that although fine-tuning for more epochs provides no additional gain in average test accuracy, it improves model performance on minority samples.

Table 5: Ablations on using different configurations for fine-tuning DeiT-III-Small model on Waterbirds dataset (Sagawa et al., 2020a). The top row corresponds to the default setting used throughout the paper for fine-tuning. The symbols ✓ and ✗ represent the corresponding setting being applied or not respectively. Best performing results are marked in **bold**.

| | Random Crop | Horizontal Flip | Mixup | CutMix | ColorJitter | Rand Augment | LabelSmoothing | Erasing | Weight Decay | Dropout | DeiT-III-Small (Pretrained on IN-21k) | | |
|---|---|---|---|---|---|---|---|---|---|---|---|---|---|
| | | | | | | | | | | | **Average** | **Worst-Group** | |
| | ✓ | ✓ | ✗ | ✗ | ✗ | ✗ | ✗ | ✗ | 0.1 | 0 | 93.58 | 76.17 | (**Default**) |
| Data Augmentation | ✗ | ✗ | ✗ | ✗ | ✗ | ✗ | ✗ | ✗ | 0.1 | 0 | 92.74 | 74.95 | |
| | ✓ | ✗ | ✗ | ✗ | ✗ | ✗ | ✗ | ✗ | 0.1 | 0 | 93.56 | 75.74 | |
| | ✗ | ✓ | ✗ | ✗ | ✗ | ✗ | ✗ | ✗ | 0.1 | 0 | 93.65 | 76.24 | |
| | ✓ | ✓ | ✓ | ✗ | ✗ | ✗ | ✗ | ✗ | 0.1 | 0 | 94.08 | 79.60 | |
| | ✓ | ✓ | ✗ | ✓ | ✗ | ✗ | ✗ | ✗ | 0.1 | 0 | 93.65 | 77.57 | |
| | ✓ | ✓ | ✓ | ✓ | ✗ | ✗ | ✗ | ✗ | 0.1 | 0 | 93.68 | 76.64 | |
| | ✓ | ✓ | ✗ | ✗ | ✓ | ✗ | ✗ | ✗ | 0.1 | 0 | 94.72 | 75.55 | |
| | ✓ | ✓ | ✗ | ✗ | ✗ | ✓ | ✗ | ✗ | 0.1 | 0 | **94.98** | **81.15** | |
| | ✓ | ✓ | ✓ | ✓ | ✓ | ✓ | ✗ | ✗ | 0.1 | 0 | 93.60 | 74.77 | |
| Regularization | ✓ | ✓ | ✗ | ✗ | ✗ | ✗ | ✓ | ✗ | 0.1 | 0 | 93.48 | 74.45 | |
| | ✓ | ✓ | ✗ | ✗ | ✗ | ✗ | ✗ | ✓ | 0.1 | 0 | 92.48 | 72.45 | |
| | ✓ | ✓ | ✗ | ✗ | ✗ | ✗ | ✗ | ✗ | 0.05 | 0 | 93.51 | 77.73 | |
| | ✓ | ✓ | ✗ | ✗ | ✗ | ✗ | ✗ | ✗ | 0.5 | 0 | 94.09 | 73.99 | |
| | ✓ | ✓ | ✗ | ✗ | ✗ | ✗ | ✗ | ✗ | 0.1 | 0.5 | 93.70 | 73.52 | |

rate as compared to the average loss. Specifically, the average train loss takes 20 epochs of fine-tuning to reach near-zero values, while training loss on minority group plateaus after 40 epochs. Similarly, we see that although the average test accuracy of the model stops increasing after 30 epochs, the accuracy of minority samples reaches a stationary state after 50 epochs of fine-tuning. These results reveal two key observations: (1) While longer fine-tuning does not benefit the average test accuracy, it plays a positive role in improving model performance on minority samples, and (2) ViT models do not overfit with longer fine-tuning.

### 5.6 How does different training configuration affect robustness to spurious correlations?

In this section, we aim to disentangle the effect of different training configurations on robustness to spurious correlations for DeiT-III (Touvron et al., 2022) models. Unlike ViT (Dosovitskiy et al., 2021), DeiT-III models

Table 6: Ablations on using different configurations for fine-tuning BiT-R50x1 model on Waterbirds dataset (Sagawa et al., 2020a). The top row corresponds to the default setting used. The symbols ✓and ✗ represent the corresponding setting being applied or not respectively. Best performing results are marked in **bold**.

| Mixup | CutMix | Auto Augment | Rand Augment | Erasing | BiT-M-R50x1 | | BiT-S-R50x1 | | |
|:---:|:---:|:---:|:---:|:---:|:---:|:---:|:---:|:---:|:---:|
| | | | | | **Average** | **Worst-Group** | **Average** | **Worst-Group** | |
| ✗ | ✗ | ✗ | ✗ | ✗ | 92.10 | 75.10 | 86.37 | 63.56 | (**Default**) |
| ✓ | ✗ | ✗ | ✗ | ✗ | 91.36 | 62.77 | 82.79 | 42.79 | |
| ✗ | ✓ | ✗ | ✗ | ✗ | 85.36 | 40.80 | 76.28 | 20.72 | |
| ✓ | ✓ | ✗ | ✗ | ✗ | 87.80 | 52.10 | 78.90 | 31.60 | |
| ✗ | ✗ | ✓ | ✗ | ✗ | **93.30** | **78.81** | **86.89** | **65.42** | |
| ✗ | ✗ | ✗ | ✓ | ✗ | 93.87 | 78.66 | 87.19 | 64.33 | |
| ✗ | ✗ | ✗ | ✗ | ✓ | 93.57 | 75.85 | 86.98 | 62.90 | |

are pre-trained using strong data augmentations. Further, for ImageNet (Deng et al., 2009) training, Touvron et al. (2021a) have shown the positive impact of different data augmentation and regularization approaches on model accuracy. Motivated by the findings, we aim to verify if applying different training schemes during fine-tuning can further improve model robustness to spurious correlations. Table 5 reports results for fine-tuning on Waterbirds (Sagawa et al., 2020a) using different configurations[1]. For this experiment, we use the DeiT-III-Small architecture pre-trained on ImageNet-21k. The top row in Table 5 corresponds to the standard fine-tuning setting we use throughout the paper for consistency and fair evaluation between different architectures. Considering data augmentation schemes, we observe that Mixup (Zhang et al., 2018), CutMix (Yun et al., 2019), and Rand Augment (Cubuk et al., 2020) provide the most significant improvement in model robustness. Specifically, using Rand Augment (Cubuk et al., 2020) during fine-tuning leads to **4.98**% improvement in worst-group accuracy. Surprisingly, we also observe that simultaneously applying too many data augmentation schemes can instead hamper the worst-group accuracy. Among regularization schemes, we observe that reducing the weight decay penalty can also provide some improvement in worst-group accuracy.

Next, we investigate the impact of different training schemes on the robustness of convolutional nets to spurious correlations. Specifically, we fine-tune the BiT-R50x1 model on the Waterbirds dataset using different data augmentation approaches. We report results in Table 6. We observe that, irrespective of the pre-training dataset, using Auto Augment (Cubuk et al., 2019) and Rand Augment (Cubuk et al., 2020) improves model robustness to spurious correlations.

## 6 Conclusion

In this paper, we investigate the robustness of ViT models when learned on datasets containing spurious associations between target labels and environmental features. Our study leads to a series of new findings, including that (1) Larger pre-training dataset and increasing model capacity improves robustness to spurious correlations. However, when the pre-training dataset is relatively small, both vision transformers and CNNs display limited robustness to spurious correlation; (2) Under spurious correlation, the self-attention mechanism in ViTs plays a crucial role in guiding the model to focus on spatial locations in an image which are essential for accurately predicting the target label. (3) Improved robustness of ViT models can be attributed to better generalization capability from the counterexamples where spurious correlations do not hold. However, when such samples become scarce ViT models tend to overfit to spurious associations. We hope that our work will inspire future research on understanding the robustness of ViT models in the presence of spurious correlation.

---

[1]Refer Table 7 (Appendix) for hyper-parameters used in different training schemes.

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

## A    Societal Impact

Our project studies the robustness and generalization of Vision Transformer (ViT) models when fine-tuned on data containing spurious associations between target labels and environmental features. Our study has positive societal impacts and will further benefit understanding the robustness of ViT models when used in domains related to fairness and AI safety, such as gender classification and medical imaging. Our study does not involve any human subjects or violation of legal compliance. We do not anticipate any potentially harmful consequence of our work. Through our study and releasing our code, we hope to raise stronger research and societal awareness towards the problem of spurious correlation and its mitigation.

## B    Implementation Details

1. **Transformers.** For ViT models, we obtain the pre-trained checkpoints from the `timm` library[2]. For downstream fine-tuning on the Waterbirds and CelebA dataset, we scale up the resolution to $384 \times 384$ by adopting 2D interpolation of the pre-trained position embeddings proposed in (Dosovitskiy et al., 2021). Note, for CMNIST we keep the resolution as $224 \times 224$ during fine-tuning. We fine-tune models using SGD with a momentum of 0.9 with an initial learning rate of 3e-2. As described in (Steiner et al., 2021), we use a fixed batch size of 512, gradient clipping at global norm 1, and a cosine decay learning rate schedule with a linear warmup.

2. **BiT.** We obtain the pre-trained checkpoints from the official repository[3]. For downstream fine-tuning, we use SGD with an initial learning rate of 0.003, momentum 0.9, and batch size 512. We fine-tune models with various capacities for 500 steps, including BiT-M-R50x1, BiT-M-R50x3, and BiT-M-R101x1.

3. **Data Augmentation Schemes.** For applying different data augmentations and regularization approaches in Section 5.6, we use the helper functions provided with `timm` library. We report different hyper-parameters used in Table 7.

Table 7: Different hyper-parameters used for data augmentation approaches in Section 5.6.

|  | Hyper-parameter Used |
|---|---|
| Mixup (Zhang et al., 2018) alpha | 0.8 |
| CutMix (Yun et al., 2019) alpha | 1.0 |
| Rand Augment (Cubuk et al., 2020) | 9/0.5 |
| ColorJitter | 0.3 |
| Horizontal Flip Prob. | 0.5 |
| Erasing Prob. | 0.25 |
| Label Smoothing | 0.1 |

## C    Extension: How does the size of pre-training dataset affect robustness to spurious correlations?

In this section, to further validate our findings on the importance of large-scale pre-training dataset, we show results on CelebA (Liu et al., 2015) dataset. We report our findings in Table 8. We also observe a similar trend for this setup that larger model capacity and more pre-training data yield a significant improvement in worst-group accuracy. Further, when pre-trained on a relatively smaller dataset such as ImageNet-1k, the performance of both transformer and CNN models are poor as compared to the ImageNet-21k counterpart.

Also, compared to BiT models, *the robustness of transformer models benefits more with a large pre-training dataset.* For example, compared to ImageNet-1k, fine-tuning DeiT-III-Base pre-trained on ImageNet-21k

---

[2] https://github.com/rwightman/pytorch-image-models/tree/master/timm
[3] https://github.com/google-research/big_transfer

Table 8: Investigating the effect of large scale pre-training on model robustness to spurious correlations when finetuned on CelebA

| | Model | Test Accuracy | |
|---|---|---|---|
| | | Average Acc. | Worst-Group Acc. |
| ImageNet-21k | DeiT-III-Base | **97.1** | **93.7** |
| | DeiT-III-Medium | 96.5 | 90.9 |
| | DeiT-III-Small | 96.2 | 88.7 |
| ImageNet-1k | DeiT-III-Base | 95.8 | 87.2 |
| | DeiT-III-Medium | 95.1 | 86.7 |
| | DeiT-III-Small | 94.8 | 85.8 |
| ImageNet-21k | BiT-M-R50x3 | 97.3 | 89.8 |
| | BiT-M-R101x1 | 97.2 | 89.8 |
| | BiT-M-R50x1 | 96.8 | 87.7 |
| ImageNet-1k | BiT-S-R50x3 | 96.4 | 88.3 |
| | BiT-S-R101x1 | 96.5 | 84.9 |
| | BiT-S-R50x1 | 96.3 | 83.1 |

improves the worst-group accuracy by **6.5%**. On the other hand, for BiT models, fine-tuning with a larger pre-trained dataset yields marginal improvement. Specifically, BiT-M-R50x3 only improves the worst-group accuracy by 1.5% with ImageNet-21k.

## D  Spurious Out-of-Distribution Detection

In this section, we study the performance of ViT models in out-of-distribution setting. Introduced in Ming et al. (2022), spurious out-of-distribution (OOD) data is defined as samples that do not contain the invariant features $\mathbf{z}^{inv}$ essential for accurate classification, but contain the spurious features $\mathbf{z}^e$. Hence, these samples are denoted as $\mathbf{x}_{ood} = \rho(\mathbf{z}^{\bar{y}}, \mathbf{z}^e)$ where $\bar{y}$ is an out-of-class label, such that $\bar{y} \notin \mathcal{Y}$. In the problem of `waterbird` vs `landbird` classification, an image of a person standing in forest would be an example of spurious OOD, since it contains different semantic class `person` $\notin \{$`waterbird`, `landbird`$\}$, yet has the environmental features of land background. A non-robust model relying on the background feature may classify such OOD data as an in-distribution class with high confidence. Hence, we aim to understand if self-attention based ViT models can mitigate this problem and if so, to what extent.

To investigate the performance of different models against spurious OOD examples, we use the setup introduced in Ming et al. (2022). Specifically, for Waterbirds (Sagawa et al., 2020a) we test on subset of images of land and water sampled from the Places dataset (Zhou et al., 2017). Considering, CelebA (Liu et al.,

Table 9: **Spurious OOD evaluation.** OOD detection performance of ViT and BiT models when finetuned on Waterbirds

| Model | Waterbirds | | CelebA | | CMNIST | |
|---|---|---|---|---|---|---|
| | FPR95↓ | AUROC↑ | FPR95↓ | AUROC↑ | FPR95↓ | AUROC↑ |
| ViT-B/16 | **56.8** | **91.0** | **60.5** | **88.4** | **7.4** | **98.8** |
| ViT-S/16 | 62.2 | 87.0 | 61.3 | 86.7 | 8.7 | 97.7 |
| ViT-Ti/16 | 79.5 | 71.6 | 94.3 | 72.7 | 16.4 | 96.7 |
| BiT-M-R50x3 | 96.0 | 59.0 | 63.8 | 85.3 | 45.9 | 84.1 |
| BiT-M-R101x1 | 95.5 | 59.5 | 70.3 | 85.6 | 44.5 | 81.4 |
| BiT-M-R50x1 | 95.1 | 63.4 | 69.7 | 85.7 | 30.0 | 88.4 |

2015) as in-distribution, our test suite consists of images of `bald male` as spurious OOD, since they contain environmental features (`gender`) without invariant features (`hair`). For CMNIST, the in-distribution data contains digits $\mathcal{Y} = \{0, 1\}$ and the background colors, $\mathcal{E} = \{$`red, green, purple, pink`$\}$. We use digits $\{5, 6, 7, 8, 9\}$ with background color `red` and `green` as test OOD samples. We report our findings in Table 9. Clearly, ViT models achieve better OOD evaluation metrics as compared to BiTs. Specifically, ViT-B/16 achieves +**32**% higher AUROC than BiT-M-R50x3, considering Waterbirds (Sagawa et al., 2020a) as in-distribution.

# E    Extension: Color Spurious Correlation

To further validate our findings beyond natural background and gender as spurious (*i.e.* environmental) features, we provide additional experimental results with the ColorMNIST dataset, where the digits are superimposed on colored backgrounds. Specifically, it contains a spurious correlation between the target label and the background color. Similar to the setup in (Ming et al., 2022), we fix the classes $\mathcal{Y} = \{0, 1\}$ and the background colors, $\mathcal{E} = \{$`red, green, purple, pink`$\}$. For this study, label $y = 0$ is spuriously correlated with background color $\{$`red`, `purple`$\}$, and similarly, label $y = 1$ has spurious associations with background color $\{$`green`, `pink`$\}$. Formally, we have $\mathbb{P}(e = $`red`$|y = 0) = \mathbb{P}(e = $`purple`$|y = 0) = \mathbb{P}(e = $`green`$|y = 1) = \mathbb{P}(e = $`pink`$|y = 1) = 0.45$ and $\mathbb{P}(e = $`green`$|y = 0) = \mathbb{P}(e = $`pink`$|y = 0) = \mathbb{P}(e = $`red`$|y = 1) = \mathbb{P}(e = $`purple`$|y = 1) = 0.05$. Note that, while fine-tuning the models, we fix the foreground color of digits as `white`.

**Results and insights on robustness performance.**   We compare model predictions on samples with the same class label but different background & foreground colors. Given a data point $(\mathbf{x}_i, y_i)$, we modify the background and foreground color of $\mathbf{x}_i$ randomly to generate a new test image $\bar{\mathbf{x}}_i$ with the constraint of having the same semantic label. In evaluation, the background color is chosen uniform-randomly from the set of colors: $\{$`#ecf02b, #f06007, #0ff5f1, #573115, #857d0f, #015c24, #ab0067, #fbb7fa, #d1ed95, #0026ff`$\}$ and the foreground color is selected randomly from the set $\{$`black`, `white`$\}$. For evaluation, we form a dataset consisting of 2100 samples. The results reported are averaged over 50 random runs. Figure 6 depicts the distribution of training samples in the CMNIST dataset (**left**) and some representative examples after transformation (**right**).

We report our findings in Figure 7. Our operating hypothesis is that a robust model should predict the same class label $\hat{f}(\mathbf{x}_i)$ and $\hat{f}(\bar{\mathbf{x}}_i)$ for a given pair $(\mathbf{x}_i, \bar{\mathbf{x}}_i)$, as they share exactly the same target label (i.e., the invariant feature is approximately the same). We can observe from Figure 7 that the best model ViT-B/16 obtains consistent predictions for 100% of image pairs. After extensive experimentation over all combinations, we find that setting the foreground color as `black` and the background as `white` caused the models to be most vulnerable. We see a significant decline in model consistency when the foreground color is set as `black` and the background as `white` (indicated as **BW**) as compared to random setup.

# F    Visualization

## F.1    Attention Map

In Figure 8, we visualize attention maps obtained from ViT-B/16 model for some samples in Waterbirds (Sagawa et al., 2020a) and CMNIST dataset. We use Attention Rollout (Abnar & Zuidema, 2020) to obtain the attention matrix. We can observe that the model successfully attends spatial locations representing invariant features while making predictions.

## F.2    The Attention Matrix of CMNIST

In the main text, we provide visualizations in which each image patch, irrespective of its spatial location, provides maximum attention to the patches representing essential cues for accurately identifying the foreground object. In Figure 9, we show visualizations for ViT-B/16 fine-tuned on the CMNIST dataset to further validate our findings.

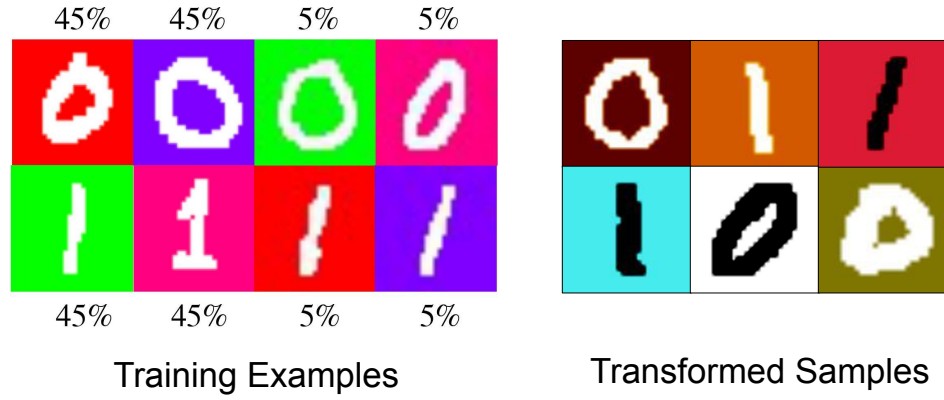

Figure 6: **CMNIST.** Distribution of training samples in CMNIST dataset(**left**) and few representative examples after transformation(**right**) as defined in Section E.

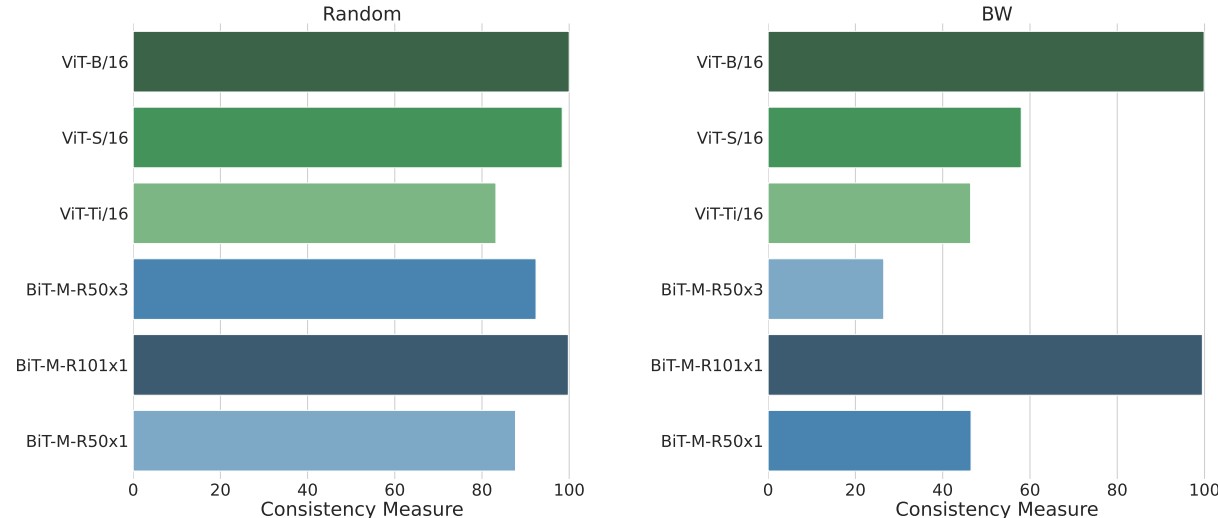

Figure 7: **Consistency Measure.** Evaluation results quantifying consistency for models of different architectures and varying capacities. We indicate the setup when the foreground color is set as `black` and the background as `white` using **BW(right)**. **Random** represents setting both the foreground and background color randomly(**left**).

## G   Software and Hardware

We run all experiments with Python 3.7.4 and PyTorch 1.9.0 using Nvidia Quadro RTX 5000 GPU.

## H   Extension: Pattern in Attention Matrix

In this section, we provide visualizations for top N patches receiving the highest attention values for ViT-Ti/16 (Figure 10) fine-tuned on Waterbirds dataset (Sagawa et al., 2020a) on various test images.

**Takeaways.**   Figure 10 shows the top N patches receiving the highest attention values for ViT-Ti/16 model fine-tuned on the Waterbirds dataset. We observe: (1) The model correctly attends to patches responsible for accurate classification in images belonging to the majority groups, i.e, `waterbird` on `water` background and `landbird` on `land` background. (2) For images belonging to minority groups (3$^{\text{rd}}$ and 4$^{\text{th}}$ row in Figure 10)

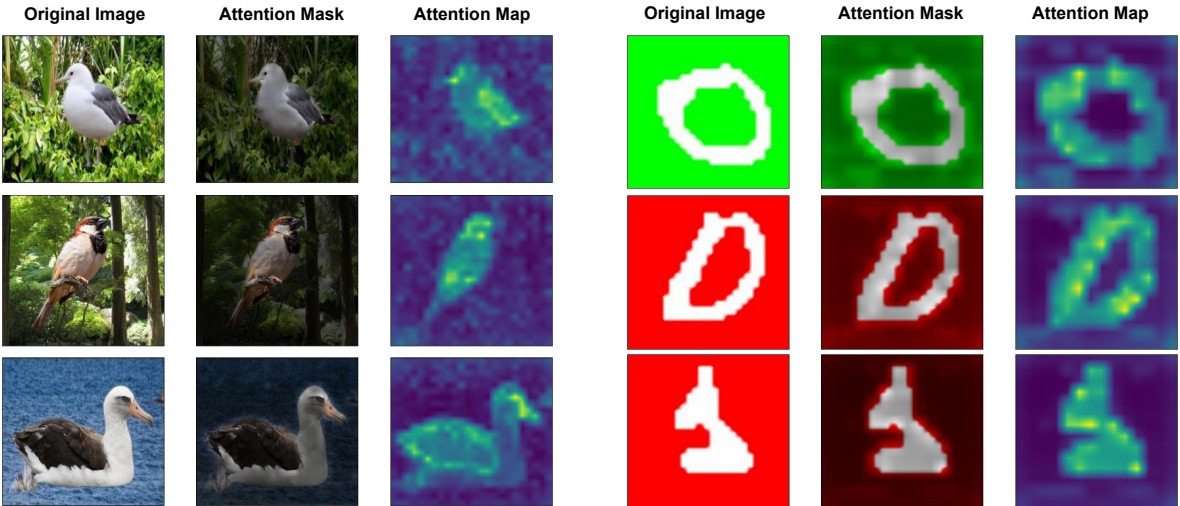

Figure 8: **Attention Map.** Visual illustration of attention map obtained from ViT-B/16 model for few representative images.

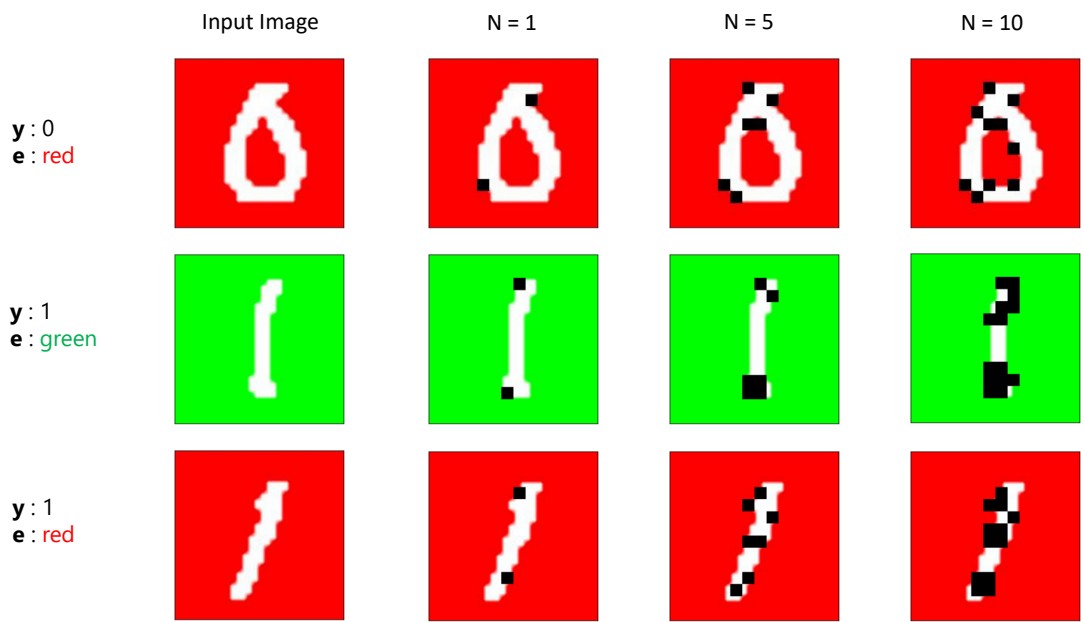

Figure 9: Visualization of the top N patches receiving the highest attention (marked in **black**) for ViT-B/16 fine-tuned on CMNIST. Investigating the attention matrix, we find that all image patches—irrespective of spatial location—provide maximum attention to the patches representing essential cues

such as `waterbird` on `land` background, the model provides maximum attention to the environmental features, exhibiting lack of robustness to spurious correlations.

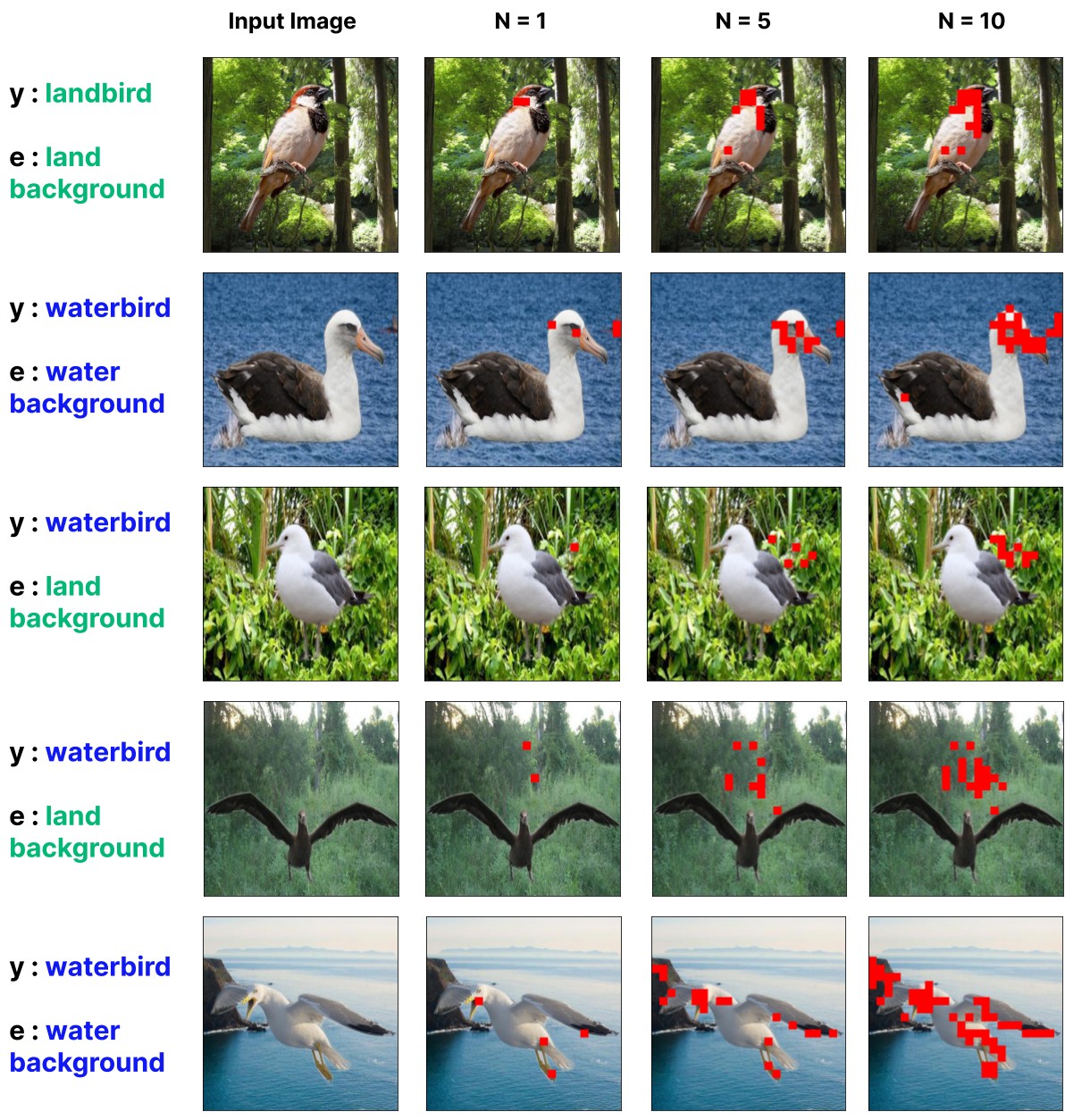

Figure 10: Visualization of the top $N$ patches receiving the highest attention (marked in red) for ViT-Ti/16 model finetuned on Waterbirds (Sagawa et al., 2020a) dataset.

## I Extension: Results highlighting importance of pre-training dataset on additional architectures

In this section, to further validate our findings on the importance of large-scale pre-training dataset we show results for additional transformer architectures: RVT (Mao et al., 2022) and PVT (Wang et al., 2021).

**Takeaways:** We report the results of finetuning different model architectures on Waterbirds (Sagawa et al., 2020a) dataset in Table 10. Note, RVT architecture is specifically modified to provide additional robustness against adversarial attacks, common corruptions, and out-of-distribution inputs. Despite these modifications, RVT models when pre-trained on ImageNet-1k perform poorly when fine-tuned on Waterbirds indicating a

Table 10: Investigating the effect of large scale pre-training on model robustness to spurious correlations when finetuned on Waterbirds

|  | Model | Params | Train Accuracy | | Test Accuracy | |
|---|---|---|---|---|---|---|
|  |  |  | Average | Worst-Group | Average | Worst-Group |
| IN-1k | RVT-Base | 85.5M | 100 | 100 | 92.7 | 66.5 |
|  | RVT-Small | 21.8M | 100 | 100 | 90.2 | 67.8 |
|  | RVT-Tiny | 8.9M | 100 | 100 | 87.8 | 48.2 |
| IN-1k | PVT-Large | 60.9M | 100 | 100 | 89.9 | 56.9 |
|  | PVT-Medium | 43.7M | 100 | 100 | 89.3 | 57.5 |
|  | PVT-Small | 23.9M | 100 | 100 | 88.8 | 49.2 |
|  | PVT-Tiny | 12.7M | 100 | 100 | 85.1 | 34.5 |

lack of robustness to spurious correlations. However, we do notice that RVT displays stronger robustness to spurious correlation than ViT when pre-trained on ImageNet-1k.

## J  Experiments on training models from scratch

In section, we train ViT and BiT models from scratch on Waterbirds(Sagawa et al., 2020a) dataset. We observe that without any pre-training, both ViT and BiT models severely overfit the training dataset to attain 100% training accuracy and significantly fail on test samples. This observation indicates that without pre-training, both transformers and CNNs have a high propensity of memorizing training samples (along with their inherent bias).

Table 11: Investigating the effect of training models on Waterbirds

|  | Model | Params | Train Accuracy | | Test Accuracy | |
|---|---|---|---|---|---|---|
|  |  |  | Average | Worst-Group | Average | Worst-Group |
| IN-21k | ViT-B/16 | 86.1M | 100 | 100 | 72.9 | 0.1 |
|  | ViT-S/16 | 21.8M | 100 | 100 | 63.9 | 0.02 |
|  | ViT-Ti/16 | 5.6M | 100 | 100 | 63.3 | 0.01 |
| IN-21k | BiT-M-R50x3 | 211M | 100 | 100 | 58.6 | 0.09 |
|  | BiT-M-R101x1 | 42.5M | 100 | 100 | 58.4 | 0.03 |
|  | BiT-M-R50x1 | 23.5M | 100 | 100 | 54.2 | 0.01 |

Table 12: Investigating the effect of applying whitening on frozen representations while linear probing pre-trained models on Waterbirds (Sagawa et al., 2020a).

|  | Model | Params | Train Accuracy | | Test Accuracy | |
|---|---|---|---|---|---|---|
|  |  |  | Average | Worst-Group | Average | Worst-Group |
| IN-21k | ViT-B/16 | 86.1M | 97.5 | 86.3 | 87.6 | 65.4 |
|  | ViT-S/16 | 21.8M | 93.5 | 32.7 | 80.5 | 28.7 |
|  | ViT-Ti/16 | 5.6M | 86.1 | 7.0 | 70.3 | 0.2 |
| IN-21k | BiT-M-R50x3 | 211M | 100 | 100 | 88.8 | 67.9 |
|  | BiT-M-R101x1 | 42.5M | 100 | 100 | 84.0 | 66.8 |
|  | BiT-M-R50x1 | 23.5M | 99.9 | 99.6 | 82.7 | 62.3 |

## K    Effect of applying whitening during linear probing

In Section 5.2 we have shown, through linear probing experiments, that simply preserving the pre-training distribution does not sufficiently provide ViT's robustness to spurious correlation. Early work by Chum & Matas (2010) noticed that co-occurrences can lead to over-counting visual patterns when comparing two vector representations. Hence, one may hypothesize that poor performance in linear probing can be due to co-occurrences in frozen embeddings. To limit the impact of co-occurrences, Jégou & Chum (2012) suggested whitening the frozen representations. In Table 12, we show the results of applying whitening while linear probing pre-trained models on Waterbirds (Sagawa et al., 2020a) dataset. We observe even after whitening the frozen embeddings, the model still fails to learn the essential cues necessary for accurate classification. This further corroborates the observation that preserving only the pre-training distribution is not sufficient for robustness to spurious correlations.

## L    Effect of transferring to smaller datasets on spurious correlations

In this section, we investigate the impact of fine-tuning models on datasets of relatively smaller sample sizes. Specifically, we reduce dataset size by subsampling the Waterbirds dataset (Sagawa et al., 2020b) with a ratio of $r = 0.25$ and $r = 0.10$. In Table 13, we report the number of samples per group for each subsampled dataset.

In Table 14, we report the results of finetuning DeiT-III (Touvron et al., 2022) and BiT (Kolesnikov et al., 2020) models on subsampled Waterbirds dataset under the different pretraining regime. We observe that: (1) Decreasing training set sample size significantly reduces model robustness to spurious correlations. Specifically, reducing the subsampling ratio from 0.25 to 0.10 results in 7.3% and 14.2% reduction in worst-group accuracy for DeiT-III and BiT-S-R50x1 respectively. (2) Large-scale pre-training improves the performance of both transformer and CNN models. Further, when pre-trained on a relatively smaller dataset such as ImageNet-1k, both transformers and convolution-based models are less robust to spurious correlations than the ImageNet-21k counterpart. Essentially, the results in Table 14 corroborate our initial observation in Section 5.1.

Table 13: Description of dataset generated by subsampling the Waterbirds (Sagawa et al., 2020a). Under-represented groups are marked in **bold**.

| Subsampling Ratio ($r$) | Total Samples | Land Bird | | Water Bird | |
|---|---|---|---|---|---|
| | | Land | Water | Land | Water |
| 1 (Original) | 4795 | 3498 | 184 | **56** | 1057 |
| 0.25 | 1199 | 874 | 47 | **13** | 265 |
| 0.10 | 480 | 352 | 20 | **5** | 103 |

## M    Studying correlation between Average and Worst-Group test accuracy

The study by (Taori et al., 2019) showed that there exists a linear relationship between the in-distribution and out-of-distribution performance of pre-trained models. Further, recent work by Andreassen et al. (2021) also provided similar findings based on several pre-trained models.

In Figure 11, we visualize the final achieved average and worst-group test accuracies at end of fine-tuning for different transformer and CNN architectures. From Figure 11, we observe for multiple models that there exists a *positive correlation* between the final achieved average and worst-group test accuracy.

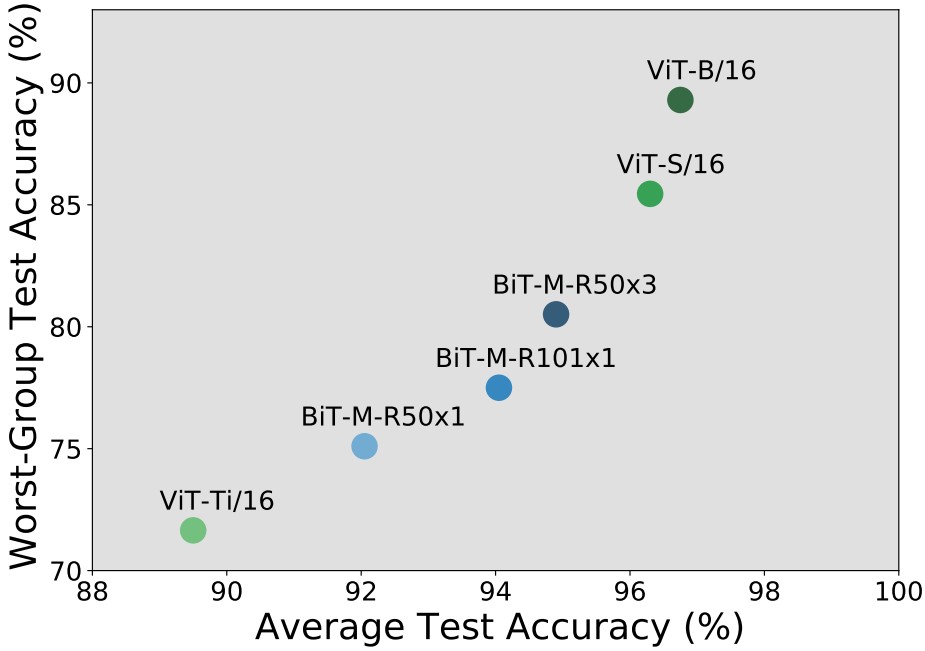

Figure 11: We investigate the correlation between average and worst-group test accuracy. We visualize the final achieved average and worst-group test accuracies for different architectures fine-tuned on Waterbirds. All models are pre-trained on ImageNet-21k.

Table 14: Investigating the effect of reducing sample size on model robustness to spurious correlations. We fine-tune DeiT-III (Touvron et al., 2022) and BiT (Kolesnikov et al., 2020) models on subsampled Waterbirds (Sagawa et al., 2020a) dataset.

| Subsampling Ratio | | Model | Params | Train Accuracy | | Test Accuracy | |
|---|---|---|---|---|---|---|---|
| | | | | Average | Worst-Group | Average | Worst-Group |
| r = 1 | IN-21k | DeiT-III-Base | 85.8M | 100 | 100 | 95.7 | 82.5 |
| | | DeiT-III-Medium | 38.3M | 100 | 100 | 94.2 | 80.8 |
| | | DeiT-III-Small | 21.8M | 100 | 100 | 93.6 | 76.2 |
| | IN-21k | BiT-M-R50x3 | 211M | 100 | 100 | 94.9 | 80.5 |
| | | BiT-M-R101x1 | 42.5M | 100 | 100 | 94.1 | 77.5 |
| | | BiT-M-R50x1 | 23.5M | 100 | 100 | 92.1 | 75.1 |
| r = 1 | IN-1k | DeiT-III-Base | 85.8M | 100 | 100 | 91.6 | 69.6 |
| | | DeiT-III-Medium | 38.3M | 100 | 100 | 93.2 | 69.8 |
| | | DeiT-III-Small | 21.8M | 100 | 100 | 90.6 | 65.1 |
| | IN-1k | BiT-S-R50x3 | 211M | 100 | 100 | 87.0 | 60.3 |
| | | BiT-S-R101x1 | 42.5M | 100 | 100 | 87.3 | 64.9 |
| | | BiT-S-R50x1 | 23.5M | 100 | 100 | 86.3 | 63.5 |
| r = 0.25 | IN-21k | DeiT-III-Base | 85.8M | 100 | 100 | 87.6 | 71.7 |
| | | DeiT-III-Medium | 38.3M | 100 | 100 | 87.9 | 67.2 |
| | | DeiT-III-Small | 21.8M | 100 | 100 | 90.1 | 63.7 |
| | IN-21k | BiT-M-R50x3 | 211M | 100 | 100 | 91.0 | 62.1 |
| | | BiT-M-R101x1 | 42.5M | 100 | 100 | 90.5 | 65.2 |
| | | BiT-M-R50x1 | 23.5M | 100 | 100 | 89.1 | 63.1 |
| r = 0.25 | IN-1k | DeiT-III-Base | 85.8M | 100 | 100 | 87.1 | 62.9 |
| | | DeiT-III-Medium | 38.3M | 100 | 100 | 83.9 | 64.1 |
| | | DeiT-III-Small | 21.8M | 100 | 100 | 84.0 | 41.4 |
| | IN-1k | BiT-S-R50x3 | 211M | 100 | 100 | 84.1 | 45.2 |
| | | BiT-S-R101x1 | 42.5M | 100 | 100 | 85.3 | 50.6 |
| | | BiT-S-R50x1 | 23.5M | 100 | 100 | 84.2 | 49.2 |
| r = 0.10 | IN-21k | DeiT-III-Base | 85.8M | 100 | 100 | 86.0 | 60.6 |
| | | DeiT-III-Medium | 38.3M | 100 | 100 | 84.8 | 58.3 |
| | | DeiT-III-Small | 21.8M | 100 | 100 | 88.0 | 54.9 |
| | IN-21k | BiT-M-R50x3 | 211M | 100 | 100 | 85.8 | 52.2 |
| | | BiT-M-R101x1 | 42.5M | 100 | 100 | 85.7 | 53.4 |
| | | BiT-M-R50x1 | 23.5M | 100 | 100 | 82.8 | 49.3 |
| r = 0.10 | IN-1k | DeiT-III-Base | 85.8M | 100 | 100 | 85.7 | 52.1 |
| | | DeiT-III-Medium | 38.3M | 100 | 100 | 80.0 | 48.1 |
| | | DeiT-III-Small | 21.8M | 100 | 100 | 81.6 | 33.8 |
| | IN-1k | BiT-S-R50x3 | 211M | 100 | 100 | 78.4 | 29.7 |
| | | BiT-S-R101x1 | 42.5M | 100 | 100 | 80.5 | 37.5 |
| | | BiT-S-R50x1 | 23.5M | 100 | 100 | 79.1 | 35.0 |

