# OpenReview forum: "Are Vision Transformers Robust to Spurious Correlations?"
_TMLR — Rejected by TMLR_

### Review · Reviewer_LEZJ · 2022-11-01

**Summary Of Contributions:**

The authors performed an empirical study on the robustness of vision transformers on spurious correlations. They investigated the generalization performance of ViT and its variants with CNNs on three datasets: Waterbirds, CelebA, and ColorMNIST. They also performed ablation studies on the influence of pre-train dataset size, sufficiency of linear probing, self-attention, data imbalance, etc. They found that vision transformers are in general more robust to spurious correlations compared to CNNs if the pre-training is sufficient.

**Audience:**

Yes

**Broader Impact Concerns:**

No ethical concerns observed.

**Claims And Evidence:**

Yes

**Requested Changes:**

- Use Swin for the experiments in Sec 4.1 & 4.2;
- Compare the computational cost of ViT and CNNs;
- Investigate the effect of transferring to smaller datasets on spurious correlations;
- Potentially investigate on medical data;


**Strengths And Weaknesses:**

Strengths:
- The authors thoroughly investigate the influencing factors of the robustness of vision transformers with respect to spurious correlations;
- The paper is overall clear and well written;

Weaknesses:
- In Figure 2, it seems there’s no significant difference between the consistency performance between ViT and BiT?
- In Sec 4.1 & 4.2, the authors didn’t include Swin transformers in the experiments. Perhaps they might outperform ViT since they are the state-of-the-art vision transformers?
- There’s no computational expense comparison (FLOPs and parameter size) between ViT and CNNs. The authors might want to discuss a bit on the trade-off between computational efficiency and performance for the readers;
- The datasets used in the paper are relatively large in sample size. It might be interesting to investigate the robustness of ViT when transferred on smaller dataset;
- The paper is primarily focused on natural images. It could be beneficial to investigate the spurious robustness of ViT on medical data. There are papers [1] in the literature on perturbation robustness of ViT on medical data but not for spurious correlations.


[1] Ghaffari Laleh, Narmin, et al. "Adversarial attacks and adversarial robustness in computational pathology." Nature Communications 13.1 (2022): 1-10.

---

> ### Author Response · Authors · 2022-11-24
> **Thank you for the constructive feedback**
>
> We thank the reviewer for the constructive feedback! We respond to your suggestions in detail below, and have incorporated the changes in our updated manuscript.
>
> > **Q1. Using Swin for the experiments in Sec 4.1 & 4.2**
>
> That's a great point. Indeed, we had investigated using advanced transformer architecture (e.g. Swin) beyond ViT. The results are summarized in **Table 3**. The setup is similar to Section 4, where all models are fine-tuned on Waterbirds, using both ImageNet-21k and ImageNet-1k in pre-training. The results can be viewed as an expansion of Table 1 in Section 4.1.
>
>
> > **Q2. Compare the computational cost of ViT and CNNs**
>
> Thanks for your suggestion. We have updated **Table 3**, with parameter size and FLOPs for both transformer and CNN models.
>
> > **Q3. Investigate the effect of transferring to smaller datasets on spurious correlations**
>
> Another great suggestion. We performed new experiments by subsampling the Waterbirds dataset with a ratio of 0.25 and 0.1. The subsampled datasets have only 13 and 5 samples in the under-represented group (Waterbird on the land background) respectively. Our results indicate that decreasing the total number of samples negatively impacts model robustness to spurious correlations. Specifically, reducing the subsampling ratio from 0.25 to 0.10 results in a 14.2% reduction in worst-group accuracy for BiT-S-R50x1.
>
> We also observe that large-scale pre-training improves the performance of both transformer and CNN models. Further, when pre-trained on a relatively smaller dataset such as ImageNet-1k, both transformers and convolution-based models are less robust to spurious correlations compared to the ImageNet-21k counterpart. These results essentially corroborate our initial observations in Section 5.1 (main paper). We have added the detailed results in **Section L** in Appendix.
>
>
> > **Q4. Potentially investigate medical data**
>
> We agree this would be really interesting. In our current evaluation, we follow the literature and use the commonly studied benchmark datasets. Our datasets cover all the tasks from [1].
> Following the findings of [2], we investigated the Chest X-ray dataset (https://nihcc.app.box.com/v/ChestXray-NIHCC) but we could not find labeling for the spurious attribute in the downloaded dataset. Further, following [3], we also studied the ISIC dataset (https://www.isic-archive.com) but again annotations for the spurious attribute were not present. Hence, we plan to investigate more on the medical data as you suggested, and include the results in the final version.
>
> [1] Shiori Sagawa, Pang Wei Koh, Tatsunori B Hashimoto, and Percy Liang. Distributionally robust neural networks for group shifts: On the importance of regularization for worst-case generalization. In ICLR, 2020.
>
> [2] Oakden-Rayner, L., Dunnmon, J., Carneiro, G. and Ré, C., 2020, April. Hidden stratification causes clinically meaningful failures in machine learning for medical imaging. In Proceedings of the ACM conference on health, inference, and learning (pp. 151-159).
>
> [3] Goel, K., Gu, A., Li, Y. and Re, C., 2020, September. Model Patching: Closing the Subgroup Performance Gap with Data Augmentation. In International Conference on Learning Representations.

---

### Review · Reviewer_8Ndd · 2022-11-04

**Summary Of Contributions:**

The paper investigates the robustness of different vision architectures (mainly CNNs and ViTs) with respect to learning spurious correlations. In particular, it investigates performance of various models on images where the context is atypical for the class: the running example is waterbirds pasted onto a watery background (in distribution) vs. waterbirds pasted onto a land-based background ("worst group" accuracy for this task), and vice-versa for land birds. Another example is classifying hair colour, with grey hair being more prevalent on males than on females in the training dataset used (CelebA).

The paper performs a study that provides evidence that ViTs are more robust against relying on spurious correlations (e.g. the background). The paper investigates the effect of pre-training scale and model size on worst-group accuracy, both increasing robustness to spurious correlations. The paper visualizes attention maps to provide intuition about which features are being used by various ViT models. Finally, the paper performs additional analyses: looking at the effect of pre-training length, and performance under "spurious OOD detection" (performance when new classes are introduced).

**Audience:**

Yes

**Broader Impact Concerns:**

I have no broader impact concerns.


**Claims And Evidence:**

No

**Requested Changes:**

Please address the above weakness. Most importantly, the correlation between in-distribution accuracy and robustness to spurious correlations, and the disentanglement of training algorithms and architectures.


**Strengths And Weaknesses:**

Strengths

- Better understanding of the behaviour and failure modes of different architectures is valuable, and such studies can provide useful insights.
- The paper provides experiments on 3 datasets and a range of different analysis metrics (accuracy under worst case, consistency, and OOD detection).
- The paper dives deeper into some causes of robustness (or lack of) such as frozen representations, finetuning length, pre-training).
- The paper is well written and easy to follow.

Weaknesses

While the study provides some interesting data, I feel that there are areas in which some more depth is required to support the main claims of the paper, and generate interest from TMLR's audience. In particular:

1) From the tables, it seems that average accuracy is strongly correlated with worst-group accuracy. It is not clear whether when changing dimension X (e.g. CNN->ViT), the robustness to spurious correlation is fully explained by increased in-distribution accuracy, or there is an additional effect of changing X. It would be useful to see plots of average acc vs. worst-group acc in order to know whether there is an additional robustness effect of, say using ViTs beyond increased accuracy overall. This is closely relates to the notion of "effective robustness" in the OOD literature.

2) There are several differences between the default training algorithms for ViTs and CNNs. For example, the data augmentation and regularization hyperparameters. It would be useful to see a disentanglement of the effect of say data augmentation and architecture (e.g. if one applied DeiT-style training to a CNN, would it also be more robust to spurious correlations?).


A couple of other things that could be improved:

1) I think the Consistency Measure defined confounds random inconsistencies in predictions with inconsistencies specifically due to the change of environment / OOD scenario. E.g. A model that makes many mistakes, but the same number of mistakes in-distribution and out-of-distribution would get a low consistency (yet the model's performance is not at all effected by the distribution shift).  I feel that the measure needs to be compared to, or normalized by, a measure of in-distribution consistency.

2) The spurious OOD detection is not quite clear to me. In these experiments, a classifier (e.g. water vs. land bird) is asked to classify an unseen class (e.g. person). It is not clear what the "correct" behaviour is in this situation. I think the assumption is that maximum entropy --- predicting 50% land 50% water bird --- would be correct, but I think this is not a reasonable assumption: given that the model cannot abstain, any prediction is equally reasonable/unreasonable. Without an "abstain" class, I feel the metric is underspecified. I appreciate that this task is not introduced in this paper, but perhaps more detail could be given about the metrics and what is expected.

Finally, the Related Work section is good, covering lots of the relevant literature. However, it would be nice to see a more in-depth discussion of how the results here relate to those from the (extensive) literature on OOD robustness. Do the results here essentially corroborate those findings in a different setting, or are there important differences? (See also the "effective robustness" point above).

---

> ### Author Response · Authors · 2022-11-24
> **Thank you for the constructive feedback**
>
> We thank the reviewer for the careful read and constructive comments! We respond to your suggestions in detail below, and have incorporated the changes in our updated manuscript.
>
> > **Q1. Correlation between in-distribution accuracy and robustness to spurious correlations**
>
> That's a great point. To investigate the correlation between in-distribution accuracy and robustness to spurious correlations we added new plot of average acc vs worst-group acc in **Section M** in the Appendix.
>
> > **Q2. Disentanglement of the effect of data augmentation and architecture**
>
> Excellent suggestion! We performed new experiments to disentangle the impact of different data augmentation and regularization approaches on robustness to spurious correlations in CNNs. Specifically, we use the BiT-R50x1 model and finetune on Waterbirds. The results indicate that, irrespective of the pre-training dataset, using data augmentation approaches such as AutoAugment [1] and RandAugment [2] during fine-tuning can improve model robustness to spurious correlations. We have added the detailed experimental setup and results in **Table 6** in the revised draft.
>
> > **Consistency Measure Suggestion**
>
> Thanks for pointing that out. To accurately capture the model's robustness to spurious correlations, we normalize the current metric by in-distribution accuracy. Accordingly, we have updated the definition of the Consistency Measure and results in the revised draft.
>
> > **Spurious OOD Clarification**
>
> As suggested, we have moved the Spurious Out-of-Distribution section from the main paper to **Section D** in Appendix.
>
> To clarify, spurious OOD samples do not contain the invariant features essential for accurate classification, but contain spurious features. So, an image of a person standing in the forest would be an example of spurious OOD, since it contains a different semantic class *person*, yet has the environmental features of *land background*. Hence, a non-robust model relying on the background feature classifies such OOD data as an in-distribution class with **high confidence**. As the reviewer noted, a robust model relying only on invariant features should output maximum entropy prediction on the spurious OOD samples.
>
> > **Additional Related Works**
>
> Thanks for the suggestion. We have extended the related works with findings of some recent works [3,4] studying the domain of *effective OOD robustness*.
>
>
> [1] Cubuk, E.D., Zoph, B., Mane, D., Vasudevan, V. and Le, Q.V., 2019. Autoaugment: Learning augmentation strategies from data. In Proceedings of the IEEE/CVF Conference on Computer Vision and Pattern Recognition (pp. 113-123).
>
> [2] Cubuk, E.D., Zoph, B., Shlens, J. and Le, Q.V., 2020. Randaugment: Practical automated data augmentation with a reduced search space. In Proceedings of the IEEE/CVF conference on computer vision and pattern recognition workshops (pp. 702-703).
>
> [3] Taori, R., Dave, A., Shankar, V., Carlini, N., Recht, B. and Schmidt, L., 2019. When robustness doesn’t promote robustness: Synthetic vs. natural distribution shifts on imagenet.
>
> [4] Andreassen, A., Bahri, Y., Neyshabur, B. and Roelofs, R., 2021. The evolution of out-of-distribution robustness throughout fine-tuning. arXiv preprint arXiv:2106.15831.

---

### Review · Reviewer_Qrs4 · 2022-11-15

**Summary Of Contributions:**

This paper systematically evaluates the impact of spurious correlations in various vision architectures. Spurious correlations are defined as patterns that appear frequently in the training set, but do not act as discriminative patterns in the test data. Therefore, a model which relies on spurious correlations may perform very well in the training set, but poorly for the test examples where the spurious correlations do not hold. The authors investigate the role of different architectures (ViT, DeiT and ResNet(BiT) variants), as well as pre-training datasets. They report their findings in Waterbirds and CELEBA datasets. They also provide many ablations to demonstrate their conclusions.

**Audience:**

Yes

**Claims And Evidence:**

Yes

**Requested Changes:**

- The impact of regularization and data augmentations should be investigated more, both in terms of pre-training and downstream fine-tuning. Please see Weaknesses.

- Linear probe experiment is interesting. There are ways (e.g. whitening[R1]) to reduce the impact co-occurences in frozen embeddings. It would be interesting to see if whitening helps with spurious correlations.

[R1] H. Jegou and O. Chum, “Negative evidences and co-occurences in image retrieval: The benefit of PCA and whitening,” in ECCV, 2012



**Strengths And Weaknesses:**

Strengths:
- The paper is mostly well written. The authors define the problem of spurious correlations very clearly.
- The authors use various different architectures for systematic evaluations.
- Discussion section covers many conclusions about spurious correlations and their causes. For each subsection, the authors validate their claims with an experiment. For example, Section 5.3 for understanding the role of self-attention mechanism and accompanying attention visualization are useful to the reader.

Weaknesses:

Although this paper covers many possibilities in terms of architectures and pre-training, there are a couple of questions that remain unanswered.

How important are data augmentation and regularization from the perspective of spurious correlations? This can be investigated from both pre-training and downstream training perspectives.  For example, DeiT performs poorly for robustness to spurious correlations compared to ViT. The authors claim that this is due to the pretraining datasets (ViT uses ImageNet-21k, DeiT uses ImageNet-1k). However, the data augmentation techniques used in the two methods are also different. Could that have any impact on the performance? Would it be possible to also report experiments with DeiT pre-trained on ImageNet-21k?

The downstream training setup is missing critical details. More specifically, it would be interesting to know the impact of the following wrt to the robustness when fine-tuning the downstream model:
- Regularization (e.g. weight decay, dropout etc). Is there a trade-off between accuracy and robustness to spurious correlations? What happens to spurious correlations if we have a larger weight decay, dropout?
- Data augmentation (e.g. RandAugment). Does adding data augmentation help with spurious correlations? If yes, which particular augmentations specifically help (e.g. cropping, color jittering, mixup etc.)?

Writing is mostly good, but the transition between Sections 3 and 4 needs to be improved. At least the key points of the experimental setup (i.e. fine-tuning from a pre-trained checkpoint etc) needs to be discussed before delving into experiments.

---

> ### Author Response · Authors · 2022-11-24
> **Thank you for the constructive feedback**
>
> We thank the reviewer for the careful read and constructive comments! We respond to your suggestions in detail below, and have incorporated the changes in our updated manuscript.
>
> > **Q1. Would it be possible to also report experiments with DeiT pre-trained on ImageNet-21k?**
>
> That's a great suggestion. We considered that too while preparing the manuscript. However, the main challenge is that we could not find ImageNet-21k pre-trained checkpoints for DeiT models in the official repository (https://github.com/facebookresearch/deit/blob/main/README_deit.md).
>
> As an alternative, we report results using recently proposed DeiT-III [1] models. As suggested by the reviewer, we performed new experiments by finetuning DeiT-III [1] models on the Waterbirds dataset. For a fair comparison, we report results for both ImageNet-1k and ImageNet-21k pre-trained models in **Table 3** (Main paper).
>
>
> > **Q2. Impact of data augmentation and regularization on robustness to spurious correction.**
>
> Excellent point! We performed new experiments to understand and disentangle the impact of different data augmentation and regularization approaches on robustness to spurious correlations. We observe that among data augmentation schemes Mixup [2], CutMix [3] and Rand Augment [4] significantly improve model robustness to spurious correlations. Specifically, using **Random Augmentation** leads to **4.98%** improvement in worst-group accuracy. We have added the new results in **Table 5** in the revised draft.
>
> > **Q3. It would be interesting to see if whitening helps when performing linear probing.**
>
> Thanks for your suggestion. We performed new experiments by applying whitening [5] on frozen embeddings during linear-probing pre-trained models on the Waterbirds dataset. We observe that even with whitening, the model still fails to learn the essential cues necessary for accurate classification. We have added the new results in **Section K** in Appendix.
>
> > **Writing suggestion**
>
> As suggested, we have revised the transition between Section 3 and Section 4, which now highlights the key points of the experimental setup. Thanks again for the careful read and constructive feedback!
>
> [1] Touvron, H., Cord, M. and Jégou, H., 2022. Deit iii: Revenge of the vit. arXiv preprint arXiv:2204.07118.
>
> [2] Zhang, H., Cisse, M., Dauphin, Y.N. and Lopez-Paz, D., 2018, February. mixup: Beyond Empirical Risk Minimization. In International Conference on Learning Representations.
>
> [3] Yun, S., Han, D., Oh, S.J., Chun, S., Choe, J. and Yoo, Y., 2019. Cutmix: Regularization strategy to train strong classifiers with localizable features. In Proceedings of the IEEE/CVF international conference on computer vision (pp. 6023-6032).
>
> [4] Cubuk, E.D., Zoph, B., Shlens, J. and Le, Q.V., 2020. Randaugment: Practical automated data augmentation with a reduced search space. In Proceedings of the IEEE/CVF conference on computer vision and pattern recognition workshops (pp. 702-703).
>
> [5] Jégou, H. and Chum, O., 2012, October. Negative evidences and co-occurences in image retrieval: The benefit of PCA and whitening. In European conference on computer vision (pp. 774-787). Springer, Berlin, Heidelberg.

---

### Review · Reviewer_8ZHZ · 2022-12-02

**Summary Of Contributions:**

[Sincere Apologies to the authors and other reviewers for the late review].

In this paper, the authors conduct an extensive empirical study to better understand which factors in the ViT training (and architecture) pipeline are responsible for their performance on spurious correlation bench-marks. The takeaway is that large scale pre-training seems to be the key factor that confers improved robustness for these ViT models.

**Audience:**

Yes

**Broader Impact Concerns:**

None.

**Claims And Evidence:**

Yes

**Requested Changes:**

I don't have any key requested changes that affect the takeaway/results of the paper. The authors can choose to decide whether they want to revise the paper to add language that addresses the previous weaknesses that I mentioned.

**Strengths And Weaknesses:**

This review is based on the updated paper and the additional experiments and changes made by the authors.

### Strengths
As previously mentioned, this paper is well-executed, clear, and comprehensive. In the revision, the authors have even gone further to address several potential weaknesses previously identified. I believe this work details important results for scholarship seeking to identify how to make a model more resilient to spurious signals in the training set.

### Weaknesses

**Definition of Spurious Correlation**\
The definition of what a spurious correlation is in this paper is vague and imprecise. A first attempt in the introduction says: "misleading heuristics imbibed within the training dataset that are correlated with majority examples but do not hold in general", and the second one says "Spurious features refer to statistically informative features that work for the majority of training examples but do not capture essential cues related to the labels."

In the definitions above, it is unclear what 'misleading', 'essential cues' are. While this comment might seem pedantic, a clear definition of what a spurious signal is, is important for adjudicating the effectiveness of an improvement or solution. As far as I can tell, the way it has been posed here suggests that the authors confirm these features to be spurious based on 'their' domain knowledge.

To the first definition: misleading to whom? Traditional ERM does not necessarily dictate which statistical correlations a model should pick and which ones it should avoid. Consequently, if the first definition is preferred, in this work, then it should be clear 'who' is adjudicating that one correlation is spurious and another is not?

Second definition: I don't think the second definition works either. Consider a dogs vs cat task where one class is 90 percent of the training data (say cats). If all the dogs (minority class) still appear say behind a green background then the model might pickup on the green vs background correlation, which will not hold for a new distribution. I think a revision of these two definitions could be something like:

1) Definition 1: a correlation is spurious if a 'domain' expert says so;
2) Definition 2: a correlation is spurious if it varies wildly from one 'domain' to another or when a distribution shifts.

This issue with the definition does not affect any of the results in the paper though.

**List of contributions and concrete takeaways from the paper**

I don't think saying that the paper is the 'first systematic study on the robustness of Vision
Transformers when learned on datasets containing spurious correlations' should be a contribution. That statement shows precedence, but is not a contribution. Instead of the current list of contributions, I would actually suggest that the authors consider summarizing the takeaways from each experimental section. For example, each subsection of Section 5 is packed with results. It might be helpful for the authors to include a single sentence takeaway from these subsections as contributions.

**Section 5.3 on analyzing the attention matrix**\
The findings here are interesting. However, some other recent results suggest that these attention weights are unidentifiable(see: On Identifiability in Transformers, and Attention is not explanation). Consequently, it is a bit difficult to try to read too much what patches the model attends to since it is possible to change these attention weights without changing the output.

**Additional Questions**\
The effect of large-scale pre-training is largely made on the basis of Table 3, which is based on ImageNet-21k. Might the benefit from pre-training on this dataset be that it already has a substantial amount of bird images in ImageNet-21k? If one pre-trains on instagram pictures, for example, perhaps the benefit would be minimal? Essentially, I am asking if the pre-training dataset of choice matters here?

---

> ### Author Response · Authors · 2022-12-03
> **Thank you for the constructive feedback**
>
> We thank the reviewer for the careful read and constructive comments! We respond to your suggestions in detail below and have incorporated the changes in our updated manuscript.
>
> > **Q1. Definition of Spurious Correlation**
>
> We thank you for raising this concern. Our wording in introduction follows the original GDRO paper [1], which states:
>
> > "...rely on spurious correlations: misleading heuristics that work for most training examples but do not always hold."
>
> While the introduction kept the definition at a high level (to facilitate readability), we rigorously define spurious correlation in **Section 3.1**.
>
> Formally, we consider a training set, $\mathcal{D}^\text{train}$, consisting of $N$ training samples: $\\{\mathbf{x}\_i, y\_i\\}^N\_{i=1}$ , where samples are drawn independently from a probability distribution: $\mathcal{P}_{X,Y}$. Here, $X\in\mathcal{X}$ is a random variable defined in the pixel space, and $Y \in \mathcal{Y} = \\{1,\ldots,K\\}$ represents its label. We further assume that the data is sampled from a set of $E$ environments  $\mathcal{E} = \\{e_1, e_2, \cdots, e_E\\}$. The training data has spurious correlations, if the input $\mathbf{x}_i$ is generated by a combination of invariant features $\mathbf{z}^{inv}\_i \in \mathbb{R}^{d\_{inv}}$, which provides essential cues for accurate classification, and environmental features $\mathbf{z}^{e}_i \in \mathbb{R}^{d_e}$ dependent on environment $e$:
>
> \begin{align*}
>     \mathbf{x}_i = \rho(\mathbf{z}^{inv}_i, \mathbf{z}_i^e).
> \end{align*}
>
> Here $\rho$ represents a function transformation from the feature space $[\mathbf{z}^{inv}_i, \mathbf{z}_i^e]^T$ to the pixel space $\mathcal{X}$.
>
> Considering the example of _waterbird_ vs _landbird_ classification, invariant features $\mathbf{z}^{inv}_i$ would refer to signals which are essential for classifying $\mathbf{x}_i$ as $y_i$, such as the feather color, presence of webbed feet, and fur texture of birds, to mention a few. Environmental features $\mathbf{z}^{e}_i$, on the other hand, are cues not essential but correlated with target label $y_i$. For example, many waterbird images are taken in water habitats, so water scenes can be considered as $\mathbf{z}^{e}_i$. Under the data model, we form groups $g = (y,e) \in  \mathcal{Y}\times  \mathcal{E}$ that are jointly determined by the label $y$ and environment $e$. **Our mathematical definition intends to describe the problem in general sense. As you said, the concrete meaning for each environment and label are instantiated in corresponding tasks, depending on the domain knowledge.**
>
>
> > **Q2. List of contributions and concrete takeaways from the paper**
>
> That's a great suggestion. Accordingly, we have updated the key contributions in the revised draft.
>
> > **Q3. Analyzing the attention matrix**
>
> Thanks for the suggestion and pointing us to [2]. In this study we provide **primary insights** on ViT’s robustness to spurious correlations by analyzing the attention matrix. We find that the attention matrix encapsulates important information about the interaction among image patches.
>
>
> > **Q4. Additional Questions on pre-training dataset**
>
> That's a great suggestion. We would be interested in knowing that too. However, we could not find checkpoints for ViT models pre-trained on Instagram images. We agree with the reviewer that the additional robustness with large-scale pre-training may be because of more training data and better diversity in the pre-training dataset (as we have already contrasted between ImageNet-1k vs. ImageNet-21k).
>
> [1] Sagawa, S., Koh, P.W., Hashimoto, T.B. and Liang, P., 2019, September. Distributionally Robust Neural Networks. In International Conference on Learning Representations.
>
> [2] Brunner, G., Liu, Y., Pascual, D., Richter, O., Ciaramita, M. and Wattenhofer, R., 2020. On Identifiability in Transformers. In 8th International Conference on Learning Representations (ICLR 2020).

---

> > ### Comment · Reviewer_8ZHZ · 2022-12-04
> > **Thanks for the clarification**
> >
> > Thanks for addressing the comments and making changes to the draft. I think the updated list of contributions now clarifies what to expect in the rest of the document. The issue regarding pre-training dataset can be addressed in future work. I think the paper already raises several interesting points and results.

---

### Author Response · Authors · 2022-11-24
**Summary of response -- thanks to all reviewers for thorough and insightful feedback**

We are encouraged that reviewers find our analysis **interesting** (R1, R2), **systematic** (R1), **thorough** (R3) and **valuable** (R2). We are equally glad that reviewers found the paper **easy to follow** and **well-written** (R1, R2, R3).

We have addressed the reviewers’ comments and concerns in individual responses to each reviewer. The reviews allowed us to strengthen our draft, and the changes made in the revised draft are summarized below:

+ [*R1*] Added new results for finetuning DeiT-III [1] models on the Waterbirds dataset starting from ImageNet-21k/1k pre-trained checkpoints in **Table 3** (main paper).
+ [*R1*] Added new results to understand the impact of different training configurations (data augmentation and regularization approaches) during fine-tuning on model robustness. The new results are added in **Table 5** (main paper).
+ [*R1*] Added new results on applying whitening on frozen embeddings during linear-probing pre-trained models in **Section K** in Appendix.
+ [*R2*] Added new plots to understand the correlation between average and worst-group acc in **Section M** in Appendix.
+ [*R2*] Added additional results to disentangle the impact of using different data augmentation schemes during fine-tuning CNNs on spuriously correlated datasets in **Table 6** (main paper).
+ [*R3*] Added parameter size and FLOPs for both transformer and CNN models in **Table 3** (main paper).
+ [*R3*] Added additional results to investigate the impact of reducing sample size on model robustness to spurious correlations in **Section L** in the Appendix.


\* In the revised manuscript, we have marked the revisions in *blue color*.

\* For brevity, we refer to reviewers **Qrs4** as *R1*, **8Ndd** as *R2*,and **LEZJ** as *R3* respectively.

[1] Touvron, H., Cord, M. and Jégou, H., 2022. Deit iii: Revenge of the vit. arXiv preprint arXiv:2204.07118.

---

### Decision · Action_Editors · 2023-01-05

**Recommendation:** Reject

**Comment:**

One remaining concern after the rebuttal period was whether larger ViTs offered larger absolute robustness or larger effective robustness. I would recommend authors to study this in further detail, and try to disentangle effective robustness from absolute robustness. Creating a plot such as Figure 11 for the results in Tables 2 and 3 could offer initial insights.

**Audience:**

Studying the differences in how CNNs and ViTs generalize should be of interest to a large section of the deep learning community. Most robustness properties such as robustness to distribution shifts and adversarial examples seem to correlate strongly with the simple validation accuracy. I think the TMLR audience would be very interested if ViTs were more robust to spurious correlations than what would be implied by their average test accuracy.

**Claims And Evidence:**

This submission studies the robustness of vision transformers (ViTs) to spurious correlations. In particular, the average test accuracy is compared with the worst-group test accuracy for several transformer based and convolutional neural networks (CNNs), when a spurious correlation is present in the training set. The authors originally claimed that ViTs are more robust to spurious correlations than CNNs, due to the observation that ViTs achieved a higher worst-group test accuracy on both the Waterbirds and CelebA datasets. However, as reviewer 8Ndd noticed, ViTs evaluated in this submission also achieved a higher average test accuracy than the CNNs that were evaluated, and it was not clear if the higher worst-group accuracy is due to higher effective robustness to spurious correlations, or simply due to higher absolute robustness due to higher average accuracy. Authors added Figure 11 as a response to reviewer 8Ndd, which showed that the higher worst-group accuracy can be explained as a simple monotonic function of the average accuracy on Waterbirds, which does not support the claim that ViTs are more robust to spurious correlations than CNNs than would be expected purely based on the average accuracy performance.

In light of this, the authors have removed the claim that ViTs are more robust than CNNs to spurious correlations, even though it was one of the main points of the original submission, and is still the question raised in the title. Authors changed the claim in the abstract to "larger models and more pre-training data significantly improve robustness to spurious correlations". However, this is still not supported by the only figure that is relevant to the claim (Figure 11), where we see that the robustness of ViTs and CNNs all seem to be determined by the same simple function of their average accuracy. It is plausible that on the Celeb-A dataset the authors' claim is supported, however this has not been demonstrated in the submission. As reviewer 8Ndd suggested, it is necessary to remove the confounding effect that larger models and better pre-training tend to lead to higher average accuracy (and the known observation that absolute robustness tends to be a monotonically increasing function of in-domain accuracy [1, 2, 3]).

I do not believe the current submission demonstrates that ViTs (or larger ViTs and longer pre-training) are more robust to spurious correlations than one might expect just based on their higher average accuracy.

[1] Radford et al. "Learning Transferable Visual Models From Natural Language Supervision"
[2] Recht et al. "Do ImageNet Classifiers Generalize to ImageNet?"
[3] Cubuk et al. "Intriguing Properties of Adversarial Examples"